# Construction of engineered RuBisCO *Kluyveromyces marxianus* for a dual microbial bioethanol production system

Dung Minh Ha-Tran[1,2,3], Rou-Yin Lai[2], Trinh Thi My Nguyen[2], Eugene Huang[4], Shou-Chen Lo[2]*, Chieh-Chen Huang [2,5]*

1 Molecular and Biological Agricultural Sciences Program, Taiwan International Graduate Program, Academia Sinica and National Chung Hsing University, Taipei, Taiwan, 2 Department of Life Sciences, National Chung Hsing University, Taichung, Taiwan, 3 Graduate Institute of Biotechnology, National Chung Hsing University, Taichung, Taiwan, 4 College of Agriculture and Natural Resources, National Chung Hsing University, Taichung, Taiwan, 5 Innovation and Development Center of Sustainable Agriculture, National Chung Hsing University, Taichung, Taiwan

* scl@dragon.nchu.edu.tw (SCL); cchuang@dragon.nchu.edu.tw (CCH)

## Abstract

Ribulose-1,5-bisphosphate carboxylase/oxygenase (RubisCO) genes play important roles in $CO_2$ fixation and redox balancing in photosynthetic bacteria. In the present study, the kefir yeast *Kluyveromyces marxianus* 4G5 was used as host for the transformation of form I and form II RubisCO genes derived from the nonsulfur purple bacterium *Rhodopseudomonas palustris* using the Promoter-based Gene Assembly and Simultaneous Overexpression (PGASO) method. *Hungateiclostridium thermocellum* ATCC 27405, a well-known bacterium for its efficient solubilization of recalcitrant lignocellulosic biomass, was used to degrade Napier grass and rice straw to generate soluble fermentable sugars. The resultant Napier grass and rice straw broths were used as growth media for the engineered *K. marxianus*. In the dual microbial system, *H. thermocellum* degraded the biomass feedstock to produce both $C_5$ and $C_6$ sugars. As the bacterium only used hexose sugars, the remaining pentose sugars could be metabolized by *K. marxianus* to produce ethanol. The transformant RubisCO *K. marxianus* strains grew well in hydrolyzed Napier grass and rice straw broths and produced bioethanol more efficiently than the wild type. Therefore, these engineered *K. marxianus* strains could be used with *H. thermocellum* in a bacterium-yeast coculture system for ethanol production directly from biomass feedstocks.

## Introduction

The thermotolerant yeast *Kluyveromyces marxianus* is a promising candidate for fuel ethanol production as it possesses several advantageous characteristics for biotechnological applications such as the faster growth rate relative to *Kluyveromyces lactis* [1] or *Saccharomyces cerevisiae* [2], the ability to assimilate a wide range of sugars [3], thermotolerance [4–6], secretion of lytic enzymes and the ability to produce ethanol at elevated temperatures [7]. Specifically, *K.*

**Data Availability Statement:** All relevant data are within the paper.

**Funding:** The design of the study and collection, analysis, and interpretation of data were supported

by the Ministry of Science and Technology (https://www.most.gov.tw/) [104-2621-M-005-003-MY3, 107-2621-M-005-007- MY3, 107-2321-B-005-004, and 108-2321-B-005-009 to CCH and 107-2621-M-005-001 to SCL.

**Competing interests:** The authors have declared that no competing interests exist.

**Abbreviations:** 3-PGA, 3-phosphoglycerate; ADHI, Alcohol dehydrogenase; DNS, 3,5-dinitrosalicylic acid; RuBisCO, Ribulose-1,5-bisphosphate carboxylase/oxygenase; GapDH, Glyceraldehyde 3-phosphate dehydrogenase; PBS, Phosphate-buffered saline; PGASO, Gene Assembly and Simultaneous Overexpression; PGK, 3-phosphoglycerate kinase; PMSF, Phenylmethylsulfonyl fluoride; PRK, Phosphoribulokinase; R5P, Ribose-5-phosphate; RuBP, Ribulose 1,5-biphosphate.

*marxianus* can grow on both hexoses and pentoses over a wide range of pH (pH 2.5–9), tolerate high temperature (up to 46˚C) and toxin (furaldehyde) [8]. The tolerance of high temperature makes *K. marxianus* a good partner for co-culturing with other cellulolytic thermophilic microorganisms as the optimal temperatures for their growth in co-culture systems are not widely different. A wide range of carbon source utilization of *K. marxianus* leverages itself when co-culturing with a robust lignocellulose-degrading bacterium such as *Hungateiclostridium thermocellum* ATCC 27405 since the bacterium can only use cellodextrins, which are glucose polymers of various lengths (G$_2$-G$_5$), as its favored carbon sources. When lignocellulosic biomass is used as a feedstock, large amounts of pentose sugars (e.g., xylose, arabinose) remaining in the culture broths that cannot be used by *H. thermocellum* can be used as substrates by *K. marxianus* to produce valuable products [9]. In addition, weak glucose repression trait makes *K. marxianus* a good choice for mixed sugar medium like lignocellulose hydrolysate [10].

Ribulose 1,5 bis-phosphate carboxylase/oxygenase (RuBisCO, EC 4.1.1.39) is the key enzyme of the Calvin-Benson-Bassham cycle (CBB cycle), which is the most important mechanism of autotrophic $CO_2$ fixation in nature [11]. In the first step of carbon fixation, RuBisCO catalyzes the addition of $CO_2$ to ribulose-1,5-bisphosphate (RuBP), which is an intermediate of the pentose phosphate pathway (PPP), to form two molecules of 3-phosphoglycerate (3-PGA). The two resultant 3-PGA molecules can be converted to two molecules each of ethanol and $CO_2$. Because of its important role, RuBisCO has been subjected to extensive studies, which included carbon assimilation to improve crop yield and investigation of the $CO_2$ fixation pathways [12–14]. In the non-sulfur purple bacterium *Rhodopseudomonas palustris*, the structural genes encoding the enzymes of the CBB cycle are organized into *cbb$_I$* and *cbb$_{II}$* operons [15]. Each operon contains genes encoding one of two distinct forms of RuBisCO. The genome of *R. palustris* consists of two active forms of RuBisCO. The form I RuBisCO includes *cbbL* (large subunit), *cbbS* (small subunit), *cbbR* (transcription regulator) and *cbbRRS* (atypical two-component systems) genes. The *cbbL* and *cbbS* genes are located at the distal end of the cbb$_I$ operon and the *cbbRRS* genes are found located between the *cbbR* and *cbbLS* genes. The form II RuBisCO comprises *cbbM* gene which is located at the distal end of the cbb$_{II}$ operon, followed by *cbbA* (fructose 1,6-bisphosphate aldolase), *cbbT* (transketolase), *cbbP* (phosphoribulokinase), and *cbbF* (fructose 1,6/sedoheptulose 1,7-bisphosphatase) genes. In the present study, the *cbbL*, *cbbS*, and *cbbP* genes were chosen to construct the form I RuBisCO cassette and the *cbbM*, *cbbP* genes were selected to construct the form II RuBisCO cassette to assemble into the K. marxianus 4G5 genome [16,17]. Form I RuBisCO is capable of fixing $CO_2$ under $CO_2$ limiting conditions and is responsible for providing cellular carbon [17]. This phenomenon is likely a response to carbon limitation and may be important for scavenging the low levels of dissolved $CO_2$ to maintain cell growth. Form I RuBisCO has much higher affinity for $CO_2$ over form II RuBisCO. The form II RuBisCO balances the intracellular redox potential under abundant carbon and electron conditions [18].

In a previous study, to enhance the $CO_2$ fixation ability of *R. palustris* DH, Du *et al.* [19] transformed the recombinant plasmid pMG-CBBM carrying form II RuBisCO gene (*cbbM*) derived from *R. palustris* NO.7 into a *R. palustris* DH strain. More recently, Guadalupe-Medina *et al.* [20] transformed two genes encoding phosphoribulokinase (PRK) (derived from *Spinacia oleracea*) and form II RuBisCO (derived from *Thiobacillus denitrificans*) into the *S. cerevisiae* genome. The results showed a 90% reduction in glycerol production and a 10% increase in ethanol production in sugar-limited chemostat cultures using a mixture of glucose and galactose feed. In the study of Li *et al.* [13], a functional carbon dioxide-fixation pathway was constructed in *S. cerevisiae* cell to improve ethanol productivity via increasing the *in-situ* $CO_2$ assimilation.

The present study sought to transform exogenous RuBisCO operons from *R. palustris* to the genome of *K. marxianus* for an improved $CO_2$ utilization and a better intracellular redox

balance. The wild-type (WT) *K. marxianus* 4G5 was used as the host cell for transforming form I RuBisCO and form II RuBisCO genes into its genome using the Promoter-based Gene Assembly and Simultaneous Overexpression (PGASO) method [21]. The inserted RuBisCO genes in the host genome were confirmed, the activities of RuBisCO genes at transcriptional and the specific enzyme activities were assessed. Moreover, the ability of the recombinant RuBisCO *K. marxianus* strains to grow on plant biomass broths or in the co-culture system with the cellulolytic bacterium *H. thermocellum* ATCC 27405 was also evaluated. Consequently, the RuBisCO *K. marxianus* strains exhibited their superiority in ethanol production over the WT. The results of this study could be applied for improving bioethanol production using plant biomass as feedstocks.

## Materials and methods

### Multiple-gene cassette construction

The PGASO method was developed to insert multiple exogenous genes into *K. marxianus* 4G5 genome [21]. Since the PGASO method was fundamentally based on site-specific homologous recombination, overhanging sequences were designed to link to the 5'-upstream sequence of a promoter, and to link to the 3'-downstream sequence of a terminator in order to facilitate an accurate gene cassette assembly into a host genome. Specifically, in the first gene cassette, a 1529 bp sequence, which is identical to the 3' region of the *K. lactis Lac4* promoter, was used as a promoter in the first gene cassette. In the last gene cassette, a 582 bp sequence, which is homologous to the 5' region of *K. lactis Lac4* promoter, was linked to the 3' end of the terminator ScTTADHI. It is noteworthy that the identities between *K. marxianus Lac4* promoter and *K. lactis Lac4* promoter, at the 5' region and at the 3' region, are 99.8% and 97.9%, respectively. The other gene cassettes were constructed to contain two parts as follows: (1) a gene sequence, at its 5' end, linked to a promoter, at its 3' end, linked to a terminator, and (2) a 55 bp overhanging sequence at the 3' end of the gene cassette that is homologous to a 5' end of the promoter of its neighboring downstream gene cassette. Individual gene fragment of *cbbL*, *cbbS*, *cbbM*, and *cbbP* from the genome of *R. palustris* CGA009 were amplified by polymerase chain reaction (PCR). The amplified gene fragments *cbbL*, *cbbS*, *cbbM*, *cbbP* and G418 (kanamycin resistance gene) were then cloned into the predesignated cassette plasmids (Table 1) with their specific independent promoter and terminator as follows: ScPGapDH-*cbbL*-ScTTGap, KlPGapDH-G418-ScTTGap, KlPADHI-*cbbS*-ScTTGap, ScPADHI-*cbbP*-ScTTADHI, and KlPPGK-*cbbM*-ScTTPGK. Subsequently, gene cassettes for PGASO technique were amplified with KOD plus DNA polymerase kit (TOYOBO Biotech) with specific primers listed in the Table 1. The cloning procedure was performed using *Escherichia coli* strain DH5α cells and Luria-Bertani (LB) medium was used as a culture medium. The antibiotic ampicillin (50 µg/mL) was used to screen the cloned plasmids. The sizes of individual transgenes and gene cassettes were shown in the Table 2. In addition, specific primers for RT-qPCR were designed to confirm the transcriptional activities of the transgenes *cbbS*, *cbbL*, *cbbM*, and *cbbP* in the host cells. Importantly, to verify the sizes and the correct orders of the gene cassettes in their predesignated assemblage in the yeast genome, the combined gene cassettes were also amplified using PCR with the forward primer of the upstream gene cassette and the reverse primer of its adjacent downstream gene cassette (Table 1).

### *K. marxianus* 4G5 transformation

For the transformation of foreign gene into *K. marxianus* genome, we followed the protocol described by Chang *et al.* [21]. Briefly, fifty µL of *K. marxianus* 4G5 strain was inoculated into a 20-mL flask having 5 mL of liquid YPD-20 medium, which contained in one liter the

**Table 1. Plasmids and primers used in the PGASO constructions and RT-qPCR experiments.**

| Plasmid | Description |
|---|---|
| pUC18 | Ampicillin resistant; multicopy plasmid with a ColE1 –type replicon |
| Cassette 1 pUC18-Kl PLac4 | pUC18 derivative with a 1529 bp portion of *Kluyveromyces lactis Lac4* promoter and *K. lactis Lac4* terminator with a 55 bp sequence at 3' end homologous to *S. cerevisiae GapDH* promoter |
| Cassette 2 pUC18-Sc PGapDH | pUC18 derivative with *S. cerevisiae GapDH* promoter and TTGap terminator with a 55 bp sequence at 3' end homologous to *S. cerevisiae PGK* promoter |
| Cassette 3 pUC18- Sc PPGK | pUC18 derivative with *S. cerevisiae PGK* promoter and TTPGK terminator with a 55 bp sequence at 3' end homologous to *K. lactis GapDH* promoter |
| Cassette 4 pUC18- Kl PGapDH | pUC18 derivative with *K. lactis GapDH* promoter *S. cerevisiae GapDH* terminator with a 55 bp overhanging sequence at its 3' region homologous to *K. lactis PGK* promoter |
| Cassette 5 pUC18-Kl PPGK | pUC18 derivative with *K. lactis PGK* promoter and *S. cerevisiae PGK* terminator with a 55 bp sequence at 3' end homologous to *K. lactis ADHI* promoter |
| Cassette 6 pUC18-Kl PADHI | pUC18 derivative with *K. lactis ADHI* promoter and *S. cerevisiae GapDH* terminator with a 55 bp sequence at 3' end homologous to *S. cerevisiae ADHI* promoter |
| Cassette 7 pUC18-Sc PADHI | pUC18 derivative with *S. cerevisiae ADHI* promoter and *ADHI* terminator with a 582 bp sequence at 3' end homologous to *K. lactis Lac4* promoter |
| Cassette 2 pUC18-Sc PGapDH-cbbL | Cassette 2 pUC18-Sc PGapDH derivative with *cbbL* |
| Cassette 6 pUC18-Kl PADHI-cbbS | Cassette 6 pUC18-Kl PADHI derivative with *cbbS* |
| Cassette 5 pUC18-Kl PPGK-cbbM | Cassette 5 pUC18-Kl PPGK derivative with *cbbM* |
| Cassette 7 pUC18-Sc PADHI-cbbP | Cassette 7 pUC18-Sc PADHI derivative with *cbbP* |
| Cassette 4 pUC18- Kl PGapDH-G418 | Cassette4 pUC18- KlPGapDH derivative with *kanR* from Lac4-KanMX cassette [21] |

| Cassette | Primer | Sequence |
|---|---|---|
| KlPLac4-KlTTLac4 | KlPLac4-3' | 5'-CCGCGGGGATCGACTCATAAAATAG-3' |
|  | KlTTLac4_ScPGapDH | 5'-CTACTATTAATTATTTACGTATTCTTTGAAATGGCAGTATTGATAATGATAAACTTATACAACATCGAAGAAGAGTCT-3' |
| ScPGapDH-*cbbL*-ScTTGapDH | ScPGapDH | 5'-AGTTTATCATTATCAATACTGCCAT-3' |
|  | ScTTGap_ScPGK | 5'-GGACTCCAGCTTTTCCATTTGCCTTGCCTGTACGGTCGTTACCATACTTGGCGGAAAAAATTCATTTGTAA-3' |
| ScPPGK-ScTTPGK | ScPGK | 5'- ACTGTAATTGCTTTTAGTTGTGTAT-3' |
|  | ScTTPGK_KlPGapDH | 5'-GGACTCCAGCTTTTCCATTTGCCTTCGCGCTTGCCTGTCGTTACCATACTAAGCTTTTTCGAAACGCAGAATTTTCG-3' |
| KlPGapDH-G418-ScTTGapDH | Kl-PGapDH | 5'-AGTATGGTAACGACCGTACAGGCAA-3' |
|  | ScTTGap_KlPGK | 5'-TACCTTTGATACCATAAAAACAAGCAAATATTCTTACTTCAAACACACCCGTGGCGGAAAAAATTCATTTGTAAACT-3' |
| KlPPGK-cbbM-ScTTPGK | KlPGK | 5'-CGGGTGTGTTGAAGTAAGAATATT -3' |
|  | ScTTPGK_KlPADHI | 5'-AGGTAAGTATGGTAACGACCGTACAGGCAAGCGCGAAGGCAAATGGAAAAGCTGGAAAGCTTTTTCGAAACGCAGAATTTT-3' |
| KlPADHI-cbbS-ScTTGapDH | KlPADHI | 5'-CCAGCTTTTCCATTTGCCTTCGCGCTTGCC-3' |
|  | ScTTGap_ScPADHI | 5'-GGAATCCCGATGTATGGGTTTGGTTGCCAGAAAAGAGGAAGTCCATATTGTACACTGGCGGAAAAAATTCATTTGTAA-3' |
| ScPADH1-cbbP-ScTTADH1 | ScPADHI | 5'-GTGTACAATATGGACTTCCTCTTTTC-3' |
|  | ScTTADH1_KlPLac4-5'End | 5'-GAAATTTAGGAATTTTAAACTTG-3' |

**Primers used for cloning**

| | | |
|---|---|---|
| Phosphoribulokinase (PRK) | cbbP-F' | 5'-CCGCCTAGGATGTCCCGTAAGCATCCG-3' |
|  | cbbP-R' | 5'-CATGCCATGGTTACTTCATGCTTCGTTTGCGGTC-3' |
| Form I RuBisCO small subunit | cbbS-F' | 5'-CCGCCTAGGATGCGACTGACCCAAGGC-3' |
|  | cbbS-R' | 5'-CATGCCATGGTCAGCCTCCGTAGCGTCG-3' |
| Form I RuBisCO large subunit | cbbL-F' | 5'-CCGCCTAGGATGAACGAAGCAGTCACC-3' |
|  | cbbL-R' | 5'-CATGCCATGGTTAGACCGAGACCGACGG-3' |

(*Continued*)

**Table 1.** (Continued)

| | | |
|---|---|---|
| Form II RuBisCO | cbbM-F' | 5'-CCGCCTAGGATGGACCAGTCGAACC-3' |
| | cbbM-R' | 5'-CCTTAATTAATTACGCCGCCTGC-3' |
| **Primers used for RT-qPCR** | | |
| Phosphorobulokinase (PRK) | r-cbbP-F' | 5'-GCCGATGAATCGATGGTGG-3' |
| | r-cbbP-R' | 5'-TTCATGCTTCGTTGCGGTC-3' |
| Form I RuBisCO small subunit | r-cbbS-F' | 5'-TGACCCAAGGCTGTTTCTCG-3' |
| | r-cbbS-R' | 5'-TTCAGTTCCATCATCACGCC-3' |
| Form I RuBisCO large subunit | r-cbbL-F' | 5'-CCGGCGTGATGGAATACAAG-3' |
| | r-cbbL-R' | 5'-ATACTTCTCCGCCGCAGTCA-3' |
| Form II RuBisCO | r-cbbM-F' | 5'-ATGATCGCCTCGTTCCTGAC-3' |
| | r-cbbM-R' | 5'-TTGATGATGGTGCCGACGAT-3' |
| Actin | 4G5 actin F | 5'-GGGCTTCGGTCAACAAAC-3' |
| | 4G5 actin R | 5'-TGGTCGGTATGGGTCAAAAGG-3' |
| **Primers to confirm gene cassettes order in *K. marxianus* genome** | | |
| Cassette (1 + 2 *cbbL*) | 1–2 F | 5'-CCGCGGGGATCGACTCATAAAATAG-3' |
| | 1–2 R | 5'-GGACTCCAGCTTTTCCATTTGCCTTCGCGCTTCGGCCTGTACGGTCGTTACCATACTTGGCGGAAAAAATTCATTTGTAA-3' |
| Cassette (2 *cbbL* + 3) | 2–3 F | 5'-AGTTTATCATTATCAATACTGCCAT-3' |
| | 2–3 R | 5'-GGACTCCAGCTTTTCCATTTGCCTTCCGCGCTTGCCTGTACGGTCGTTACCATACTAAGCTTTTTCGAAACGCAGAATTTTCG-3' |
| Cassette (3 + 4 G418) | 3–4 F | 5'- ACTGTAATTGCTTTAGTTGTGTAT-3' |
| | 3–4 R | 5'-TACCTTTGATACCATAAAACAAGCAAATATTCTTACTTCAAACACACCCGTGGCGGAAAAAATTCATTTGTAAACT-3' |
| Cassette (4 G418 + 5) | 4–5 F | 5'-AGTATGGTAACGACCGTACAGGCAA-3' |
| | 4–5 R | 5'-AGGTAAGTATGGTAACGACCGTACAGGCAAGCGCGAAGGCAAATGGAAAAGCTGGAAAGCTTTTTCGAAACGCAGAATTTT-3' |
| Cassette (5 + 6 *cbbS*) | 5–6 F | 5'-CGGGTGTGTTTGAAGTAAGAATATT -3' |
| | 5–6 R | 5'-GGAATCCCGATGTATGGGTTTGGTTGCCAGAAAAGAGGAAGTCCATATTGTACACTGGCGGAAAAAATTCATTTGTAA-3' |
| Cassette (6 *cbbS* + 7 *cbbP*) | 6–7 F | 5'-CCAGCTTTTCCATTTGCCTTCGCGCTTGCC-3' |
| | 6–7 R | 5'-GAAATTTAGGAATTTTAAACTTG-3' |

Table 2. Sizes of individual RuBisCO genes, gene cassettes used in the PGASO constructions.

| Cassette | Promoter-Terminator (bp) | RubisCO gene (bp) | Promoter-Gene-Terminator (bp) | Primers |
|---|---|---|---|---|
| 1 | 2347 | - | KlPLac4- KlTTLac4 (2347) | F-KlPLac4-3' |
| | | | | R-KlTTLac4_ScPGapDH |
| 2 | 1102 | *cbbL* (1457) | ScPGapDH-*cbbL*-ScTTGap (2559) | F-ScPGapDH |
| | | | | R-ScTTGap_ScPGK |
| 3 | 1294 | - | ScPPGK-ScTTPGK (1294) | F-ScPGK |
| | | | | R-ScTTPGK_KlPGapDH |
| 4 | 1145 | G418 (810) | KlPGapDH-G418-ScTTGap (1955) | F-Kl-PGapDH |
| | | | | R-ScTTGap_KlPGK |
| 5 | 1291 | *cbbM* (1386) | KlPPGK-cbbM-ScTTPGK (2677) | F-KlPGK |
| | | | | R-ScTTPGK_KlPADHI |
| 6 | 1301 | *cbbS* (425) | KlPADHI-cbbS-ScTTGap (1726) | F-KlPADHI |
| | | | | R-ScTTGap_ScPADHI |
| 7 | 2068 | *cbbP* (876) | ScPADHI-cbbP-ScTTADHI (2944) | F-ScPADHI |
| | | | | R-ScTTADHI_KlPLac4-5'End |

following: 10 g of yeast extract (MDBio, Inc, Taiwan), 20 g of peptone (BD), and 20 g of glucose (Showa, Japan). The flask was capped with foam stopper and incubated at 30°C and 200 rpm for 12–16 h. The target gene cassettes in a 5 μL volume with an equal molar ratio of each fragment were mixed with 40 μL of competent cells. The electroporation was carried out (1 kV, 400 Ω, 25 μF) using Gene Pluser Xcell TM Electroporation system (Bio-Rad, USA) with an aluminum cuvette (2 mm). The cells were spread onto YNBD (6.7 g of yeast nitrogen base without amino acids (Difco, USA) and 10 g of glucose in 1L) agar plates containing G418 (200 μg/mL). In the following culture experiments, YPD medium without glucose was called YPD-0, YPD medium with 8 g/L glucose was called YPD-8 and the standard YPD with 20 g/L glucose was named YPD-20.

## Plant biomass substrates

One-year old Napier grass cultivated in Taiwan was used in this study and in our previous study, the composition of dried Napier grass contains 38% hemicellulose, 44% cellulose and 8% lignin [22]. Fresh Napier grass was collected from the field, and then dried in a Benchtop shaking incubator TS-1450 (Florida 33130, USA) at 65°C for 7 days. The leaves and stems were separately ground into a fine powder using a RT-N08 pulverizing machine (Taichung City, Taiwan) and then sieved through a 420-μm screen. The powders obtained from stems and leaves were mixed in the ratio of 1:1 (w/w) before the mixture was used in the experiments. Serum bottles were loaded with the substrates and autoclaved at 121°C for 20 min. Rice straw of TNG67 variety was kindly provided by Professor Chang-Sheng Wang, Department of Agronomy, National Chung Hsing University, Taiwan. According to Amnuaycheewa *et al*. [23], rice straw comprises 34.6% cellulose, 29.7% hemicellulose, 15.3% lignin and 10.0% ash. Rice straw was dried in the Benchtop shaking incubator TS-1450 (Florida 33130, USA) at 65°C for 3 days and then grown into fine powder. The powder was sieved through a 420-μm screen before use.

## Culture medium

One liter of modified GS-2 medium contained 1.5 g $KH_2PO_4$, 2.9 g $K_2HPO_4$, 3 g sodium citrate tribasic dihydrate ($Na_3C_6H_5O_7$), 2.1 g urea, 5 g 3-(N-morpholino) propanesulfonic acid

(MOPS), 2 mg resazurin, 1 g L-cysteine. A 100 mL of 10-fold trace salt solution contained 26 mg $MgCl_2$, 11.3 mg $CaCl_2$, 0.125 mg $FeSO_4 \cdot 7H_2O$. A 100 mL of 100-fold vitamin solution contained 2 mg pyridoxine hydrochloride, 0.2 mg biotin, 0.4 mg p-aminobenzoic acid, and 0.2 mg vitamin B12 [24]. All solutions were prepared with distilled water obtained with an EcoQ Combo (LionBio, Taiwan). Trace salt and vitamin solutions were sterilized using 0.22-μm filter (StarTech, Taiwan). The pH of the GS-2 medium was adjusted to 7.2 using 5M NaOH and was purged extensively with pure nitrogen gas to create an anaerobic environment. Finally, GS-2 medium was autoclaved at 121˚C for 20 min. To maintain an anaerobic environment during fermentation process, serum bottles were strictly sealed with butyl rubber stoppers and aluminum seals. The sterilized medium GS-2 was cooled to 50˚C and aseptically supplemented with 1% (v/v) trace salt solution and 1% (v/v) vitamin solution.

## Biomass fermentation broths

For inoculum preparation, *H. thermocellum* ATCC 27405 was grown in modified GS-2 medium supplemented with cellobiose (5 g/L) until the stationary phase ($OD_{660}$ ~ 0.7) and then inoculated 1% (v/v) to modified GS-2 medium containing 100 g/L Napier grass or 100 g/L rice straw powder. Cells were grown in batch culture at 60˚C in 250-mL serum bottles containing 100 mL of the modified GS-2 medium. At the end of the fermentation (6[th] day), the fermentation broths were filtered through filter paper (6 μm) (Advantec® No. 1 90 mm, Japan), and then centrifuged at 12,800 rpm for 10 min (Hermle Z326K, Germany) to remove the remaining substrate. The pH of the culture broth was adjusted to 7.0 with 2 M NaOH and the culture broth was sterilized using 0.22-μm filter (StarTech, Taiwan) before the addition of the *K. marxianus* inocula. Seed inocula of WT, form I RuBisCO and form II RuBisCO were prepared by culturing in YPD-8 medium (with 8 g/L glucose) for 24 h to reach $OD_{660}$ ~ 1.1. One mL each of yeast culture was collected, centrifuged at 12,000 x g for 10 min at 4˚C using a Hitachi Tabletop centrifuge CT15RE (Hitachi Koki Co., Ltd, Japan) to remove the supernatant. The pellet was washed twice with phosphate-buffered saline (PBS) solution to remove the background and then inoculated 1% (v/v) into Napier grass or rice straw medium. One liter of PBS solution contained 8 g NaCl, 0.2 g KCl, 1.44 g $Na_2HPO_4$, 0.24 g $KH_2PO_4$, 0.133 g $CaCl_2.2H_2O$, and 0.1 g $MgCl_2.6H_2O$. The semi-anaerobic batch culture was performed in a 250 mL-serum bottle with 100 mL Napier grass or rice straw broth at 37˚C for 48 h.

## Co-culture fermentation

*H. thermocellum* was grown in modified GS-2 medium supplemented with cellobiose (5 g/L) until the stationary phase was reached ($OD_{660}$ ~ 0.7) and then inoculated 1% (v/v) into GS-2 medium containing 100 g/L Napier grass or 100 g/L rice straw powder. Cells were grown in batch culture at 60˚C in 250-mL serum bottles containing 100 mL of the modified GS-2 medium. This was the first step for plant biomass solubilization and bioconversion using *H. thermocellum* ATCC 27405. After 144 h, the culture stopped producing $H_2$ and $CO_2$, indicating the end of metabolic activities. One day prior to the end of the *H. thermocellum culture*, seed inocula of *K. marxianus* strains WT, form I RuBisCO and form II RuBisCO were prepared by culturing in YPD-8 medium (with 8 g/L glucose) for 24 h to reach $OD_{660}$ ~ 1.1. One mL each of the yeast cultures was collected, centrifuged at 12,800 x g for 10 min at 4˚C using a Hitachi Tabletop centrifuge CT15RE (Hitachi Koki Co., Ltd, Japan) to remove the supernatant. The pellets were washed twice with phosphate-buffered saline (PBS) solution to remove the residual media components and then inoculated 1% (v/v) to Napier grass- or rice straw-containing serum bottles for the co-culture study. The temperature of co-culture system was decreased to 37˚C to suit the growth temperature of *K. marxianus*. The co-culture process

lasted for 48 h when 1 mL each of the fermentation broth was collected, centrifuged 12,800 x g for 10 min at 4˚C, filtered with 0.22-μm filter (StarTech, Taiwan). The final ethanol concentration was measured by GC at the end of the fermentation process, $CO_2$ production was measured every 24 h using GC Agilent 7890A (Agilent Technologies, USA).

## Ethanol and gas measurement

The major fermentation end-products were determined by the GC Agilent 7890A equipped with a J&W 122–3232: 30 m x 250 μm x 0.25 μm DB-FFAP column, with a flame ionization detector (FID) for ethanol and a thermal conductivity detector (TCD) for $CO_2$, with nitrogen as the carrier gas at a flow rate of 30 mL/min. For ethanol measurement, the front inlet was used, the detector was kept at 225˚C and the oven operated from 50˚C to 150˚C, 50–100˚C at a ramp rate of 30˚C/min, and 100–150˚C at a ramp rate of 20˚C/min. For $H_2$ and $CO_2$ measurements, the back inlet was used, the detector was kept at 225˚C and the oven operated isothermally at 50˚C. In the calculations of the results, the weight of 1.84 $mg/cm^3$ of $CO_2$ at 25˚C under standard atmospheric pressure and its molecular weight 44 g/mol were used.

## Reducing sugar and protein measurement

The total reducing sugars remaining in biomass broths were measured using the 3,5-dinitrosalicylic acid (DNS) method (Miller, 1959) [25] in which the standard curve was prepared using 5-fold serial dilutions of a pure chemical sample. Protein in the broth culture supernatant was determined by the Bradford Assay with Coomassie Brilliant Blue G-250 (Merck KGaA, USA) and Bovine Serum Albumin (BSA) (Merck KGaA, USA) as the standard.

## RT-qPCR analysis

The WT and RuBisCO strains were cultured in the YPD-8 medium for 12 h at 30˚C, 200 rpm. One mL of each culture broth was collected, centrifuged at 5,000 rpm for 1 min. The supernatant was discarded, and 100 mL of the fresh YPD-8 medium was added to the Eppendorf tube to suspend the pellet. The tube was quickly immersed in liquid nitrogen for 3 s, and then transferred to the 37˚C water bath B206-T1 (Firstek, Taiwan). This step was repeated 3 times to effectively break down the cell wall. One mL of TRIzol (R) Reagent was added to the Eppendorf tube and the total mRNA was **isolated** using RNAqueous™ Total RNA Isolation Kit (Thermo Fisher Scientific, USA) according to the instruction manual. DNA-free™ kit (Ambion, Thermo Fisher Scientific, USA) was used to remove the contaminating DNA. Purified RNA was quantified at $A_{260}/A_{280}$ ratio using a Nano-100 Micro Spectrophotometer (Medclub Scientific Co., LTD, Taiwan). cDNA was synthesized using ImProm-IITM Reverse Transcription System (Promega, USA). RT-qPCR was performed using Corbett Research Rotor-Gene 3000 (Mortlake 2137, Australia) with FastStart Universal SYBR(R) Green Master (ROX) kit (Roche, USA). The thermal cycling program consisted of an activation step (95˚C, 10 min) followed by 40 cycles of denaturing (95˚C, 10 sec), annealing (60˚C, 10 sec). The relative expression levels of the RuBisCO genes were normalized to that of the Actin gene and calculated using the $2^{-\Delta\Delta Ct}$ method [26].

## Activity assay of RubisCO

RuBisCO activity measurement was performed based on the continuous spectrophotometric rate determination method described in [14,27] with some modifications. The crude cell lysate was prepared using the glass bead lysis as described by Mukherjee et al. [28]. Briefly, the yeast strains were grown semi-anaerobically in 250-mL serum bottles containing YPD-8 medium

for 12 h. Cells were harvested by centrifuging at 12,800 x g, 10 min at 4˚C, washed twice with sterile distilled water, and then resuspended in 350 μL of lysis buffer containing 50 mM Tris-HCl buffer, pH 7.5, with 2 mM EDTA, 0.1 mM phenylmethylsulfonyl fluoride (PMSF), 0.1 mM DTT, 3 mM $MgCl_2$, 10 mM NaCl. The cells in the glass bead tube were vortexed in a cyclomixer for 4 times at 2,500 rpm for 15 s each, immersing the cell suspension in ice for 15 s between 2 vortexing cycles. The cell suspension was centrifuged at 5,000 x g for 10 min at 4˚C to collect the supernatant and the supernatant was used as the crude cell lysate for enzymatic assay. The reaction mixture containing cell lysate, 50 mM HEPES buffer (pH 8), 1 mM Ribulose 1,5-bisphosphate (RuBP), 20 mM $MgCl_2$, 5 mM DTT, 20 mM $NaHCO_3$, 5 mM ATP, 0.15 mM NADH, 5 U/mL 3-phosphoglycerate kinase, 6 U Glyceraldehyde 3-phosphate dehydrogenase was prepared. Specifically, 20 μL cell lysate was added to 180 μL of reaction mixture. To initiate the reaction, 7 μL of 33 mM RuBP was added to the reaction mixture to reach a final concentration of 1 mM. The absorbance at 340 nm ($A_{340}$) was then recorded every 20 s for 30 min using Beckman Coulter—PARADIGM™ microplate reader (Beckman Coulter, USA) and the rate of decrease in $A_{340}$ was then converted to the rate of NADH oxidation. RuBisCO activity was calculated from the rate of NADH oxidation. The millimolar extinction coefficient of NADH at 340 nm was used to determine the rate of $NAD^+$ production. Total protein concentration in the cell-free extracts was measured by the Bradford protein assay (Bio-Rad Protein Assay).

Formula for RuBisCO enzyme activity calculation:

$$nmol\ min^{-1}\ mg\ protein^{-1} = \frac{(\Delta A_{340}/min\ \text{Test} - \Delta A_{340}/min\ \text{Blank})\text{x Total volume of assay (mL)}}{2\text{x}6.22\text{x}0.6\text{x mg of lysate used (mg)}}$$

$$\Delta A_{340}:\ \text{Initial } A_{340} - \text{Final } A_{340}$$

2: 2 μmoles of β-NADH are oxidized for each μmole of D-Ribulose 1,5-diphosphate used.
6.22: Millimolar extinction coefficient ($mmol^{-1}\ cm^{-1}$) for β-NADH at 340 nm.
0.6: Optical path length (cm) in an ELISA well with 200 μL reaction volume.

## Statistical analysis

Analysis of variance (Anova) was performed in ethanol and gas measurement to verify the differences between *K. marxianus* WT and RuBisCO strains. The probability was set at $p < 0.05$ and each experiment was repeated three times. All data analyses and most of graphics were performed using R software (ver. 3.6.1). Some other graphics such as representative fermentation procedures were performed using CorelDRAW X7 version (Corel Corporation).

## Results and discussion

### Construction of form I and form II RuBisCO gene cassettes

Gene cassettes were designed using the method described in Materials and Methods. Each gene cassette contained a specific promoter and a terminator. Specifically, the form I RuBisCO *cbbL* gene fragment (2559 bp) was linked with promoter GapDH (glyceraldehyde 3-phosphate dehydrogenase gene from *S. cerevisiae*) in the gene cassette 2 ScPGapDH. The selection gene neomycin phosphotransferase G418 (1955 bp) essential for kanamycin resistance was linked with promoter GapDH (glyceraldehyde 3-phosphate dehydrogenase gene from *K. lactis*) in the gene cassette 4 KlPGapDH. Form I RuBisCO *cbbS* gene fragment (1726 bp) was linked with promoter ADHI (alcohol dehydrogenase gene from *K. lactis*) in the gene cassette 6 KlPADHI and gene encoding phosphoribulokinase (*cbbP*) gene fragment (2944 bp) was driven by

promoter ADHI (alcohol dehydrogenase gene from *S. cerevisiae*) in the cassette 7 (Fig 1A). The PCR products of gene cassettes in the form I RuBisCO construction (Left panel) and the combined amplified gene cassettes (Right panel) were displayed in Fig 1B. Similarly, gene G418 fragment (1955 bp) was linked with promoter GapDH from *K. lactis* in the gene cassette 4 KlPGapDH and form II RuBisCO *cbbM* gene fragment (2267 bp) was driven by promoter PGK from *K. lactis* in the gene cassette 5 KlPGK. Finally, the phosphoribulokinase *cbbP* gene fragment (2948 bp) was driven by promoter ADHI from *S. cerevisiae* in the cassette 7 (Fig 1C). It should be noted that all the promoters used in the present study had approximately 40–55% sequence identity between each other in their 5' end regions. In PGASO method, the low sequence identity of promoters is of importance in preventing the unexpected recombination events where multi-genes could be integrated into similar regions in the host genome. The PCR products of gene cassettes in the form II RuBisCO construction (Left panel) and the combined gene cassette fragments (right panel) were displayed in Fig 1D.

## Relative gene expression levels of RuBisCO genes in the engineered *K. marxianus* strains

After the incorporation of the form I RuBisCO and form II RuBisCO gene cassettes into *K. marxianus* genome, the two recombinant *K. marxianus* strains were cultured in YPD-8 to evaluate their function via transcriptional patterns (Fig 2).

The transcriptional levels of RuBisCO genes in the engineered RuBisCO strains grown in YPD-8 medium at 12 h were evaluated by RT-qPCR (Fig 2). The results showed that, in the engineered form I RuBisCO strain, the transcriptional level of *cbbS* gene was the highest (Fig 2A). The promoter KlPADHI, which drives the *cbbS* gene was a glucose-inducible promoter as described in the study of Mazzoni et al. [29], exhibited its potent function in regulating *cbbS* gene activity. In their study, Mazzoni and colleagues found that the alcohol dehydrogenase 1 (*KlADH1*) gene in *K. lactis* was preferentially expressed in glucose-grown cells in respect of ethanol-grown cells. The low expression of *cbbL* gene, which was driven by the ScPGapDH promoter, is worthy of further examination since the reason behind the extreme low transcriptional level of this gene remains unknown. Based on the considerations from our previous study [30], gene cassette arrangements within the PGASO constructions and appropriate promoter strength probably resulted in the improvement of gene expression levels, and subsequently reduced the cell burden and the competition for transcription factors. Therefore, an arrangement of gene cassettes and/or a new, stronger promoter should be taken into consideration to enhance the expression of the *cbbL* gene in further study. In the meantime, the *cbbP* gene remained the moderate expression level in the form I RuBisCO strain. The transcriptional patterns of *cbbM* and *cbbP* genes in the form II RuBisCO strain were maintained at elevated levels in the YPD-8 medium (Fig 2B). As the key enzyme of the reductive pentose phosphate cycle, the *cbbP* gene encoding PRK enzyme is responsible for the conversion of ribose-5-phosphate (R5P) to RuBP. RuBisCO, in turn, uses RuBP as its substrate to form two molecules of 3-PGA. The results confirm that the RuBisCO genes were properly inserted into the host genome and function well in both form I and form II RuBisCO yeast strains.

## Specific enzyme activity of RuBisCO

The RuBisCO assay was performed to evaluate the functions of RuBisCO gene cassettes in the engineered strains at translational level. The data displayed in Fig 3 revealed that RuBisCO enzymes were appropriately synthesized in the engineered RuBisCO strains and these enzymes functioned properly in converting RuBP to 3-PGA, the first major step of a $CO_2$ fixation. The specific enzyme activities of RuBisCO in the present study, however, were quite low as compared

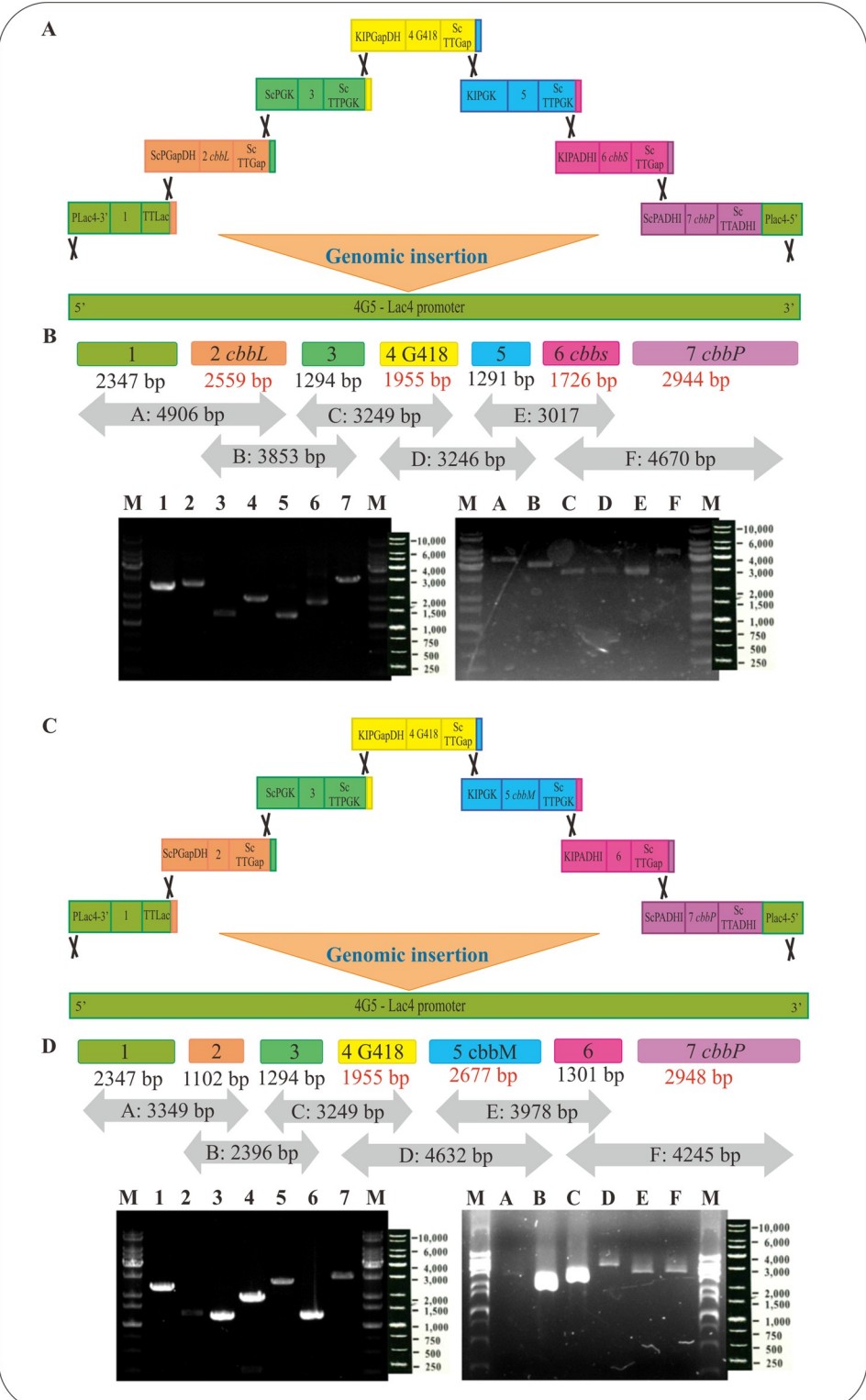

**Fig 1. The constructions of form I and form II RuBisCO, PCR products of individual gene cassette and the combined gene cassettes.** (A) The designated gene cassettes of form I RuBisCO genes. Each of the four gene cassettes contains its independent promoter, a gene fragment coding region, a terminator, and a 46 bp fragment at the 3' end of the gene cassette that is homologous to its neighboring gene cassette. After the electroporation, the gene cassettes are assembled in the predesignated order. (B) PCR products of RuBisCO gene cassettes in form I RuBisCO (left panel)

were shown as follows: M, DNA ladder; 1, Cassette 1 (2347 bp); 2, Cassette 2-*cbbL* (2559 bp); 3, Cassette 3 (1294 bp); 4, Cassette 4-G418 (1955 bp); 5, Cassette 5 (1291 bp); 6, Cassette 6-*cbbS* (1726 bp); 7, Cassette 7-*cbbP* (2944 bp). PCR products of the combined gene cassettes (right panel) were shown as follows: M, DNA ladder; A, (cassette 1 + cassette 2 *cbbL*) (4906 bp); B, (cassette 2 *cbbL* + cassette 3) (3853 bp); C, (cassette 3 + cassette 4 G418) (3249 bp); D, (cassette 4 G418 + cassette 5) (3246 bp); E, (cassette 5 *cbbM* + cassette 6 *cbbS*) (3017 bp); F, (cassette 6 *cbbS* + cassette 7 *cbbP*) (4670 bp); M, DNA ladder. (C) The designated gene cassettes of form II RuBisCO genes. Each of the three gene cassettes possesses an independent promoter, a gene fragment, a terminator, and a 46 bp fragment homologous to its neighboring cassette. After the electroporation, the gene cassettes are assembled in the predesignated order. (D) PCR products of RuBisCO gene cassettes in form II RuBisCO (left panel) were shown as follows: M, DNA ladder; 1, Cassette 1 (2347 bp); 2, Cassette 2 (1102 bp); 3, Cassette 3 (1294 bp); 4, Cassette 4-G418 (1955 bp); 5, Cassette 5-*cbbM* (2677 bp); 6, Cassette 6 (1301 bp); 7, Cassette 7-*cbbP* (2948 bp). PCR products of the combined gene cassettes (right panel) were shown as follows: M, DNA ladder; A, (cassette 1 + cassette 2) (3349 bp); B, (cassette 2 + cassette 3) (2396 bp); C, (cassette 3 + cassette 4 G418) (3249 bp); D, (cassette 4 G418 + cassette 5 *cbbM*) (4632 bp); E, (cassette 5 *cbbM* + cassette 6) (3978 bp); F, (cassette 6 + cassette 7 *cbbP*) (4245 bp); M, DNA ladder.

to the previous study [14]. This might be due to the lack of purification step in cell lysis procedure that prevented us from obtaining the best possible yield and purity of enzymes. However, the results of RuBisCO assay were in accordance with the lower accumulated $CO_2$ levels released from the form I and form II RuBisCO strains as compared to those released from the WT. These results also explained the higher ethanol titers of the engineered strains relative to the WT.

## Growth profiles and ethanol production of *K. marxianus* on glucose-free medium YPD-0 and on high glucose concentration medium YPD-20

As shown in the Fig 4A and 4B, in the YPD-0 medium where no additional carbon source was supplemented into the medium, all *K. marxianus* strains exhibited the poor growth and low ethanol yield. However, ethanol produced by engineered RuBisCO strain was significantly greater than that of the WT. In contrast, in the YPD-20 medium, all strains of *K. marxianus* grew fast and produced the highest ethanol concentrations compared to those in other media (Fig 4C and 4D). The positive correlation between fermentable sugar and ethanol concentration was obviously observed. After 24 h of fermentation, most glucose was consumed as shown

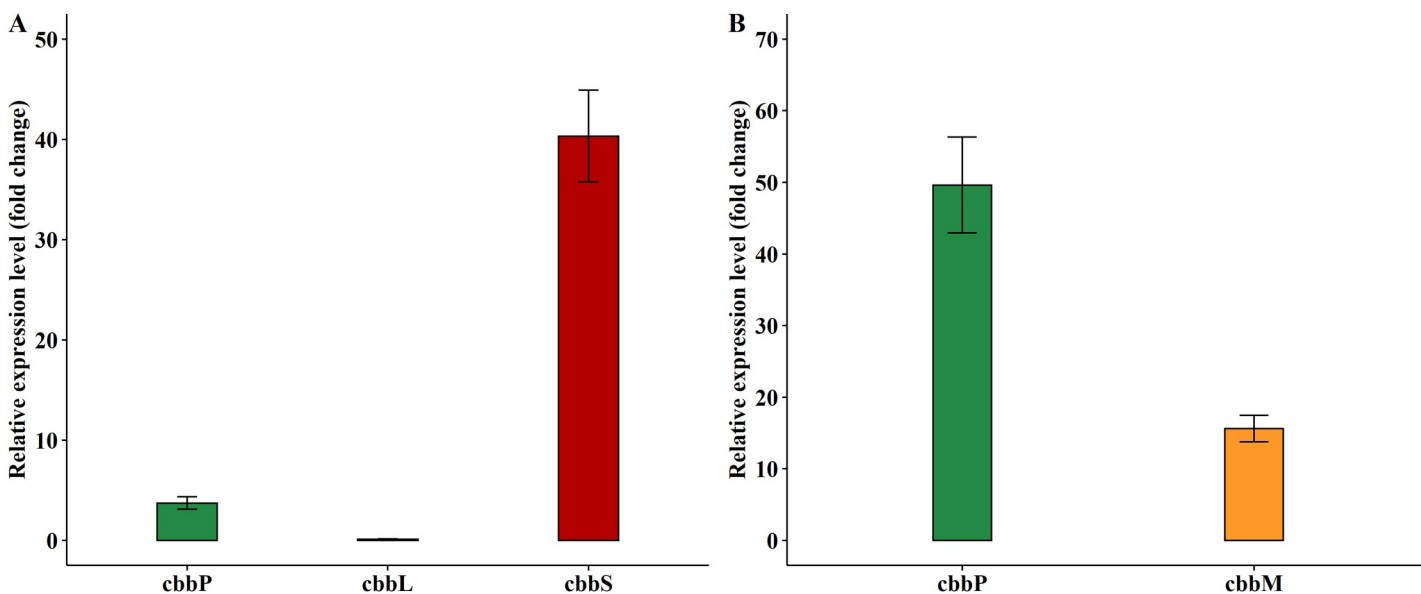

**Fig 2. RT-qPCR results.** (A) Relative transcriptional levels of RuBisCO genes in the engineered form I RuBisCO strain. (B) Relative transcriptional levels of RuBisCO genes in the engineered form II RuBisCO strain.

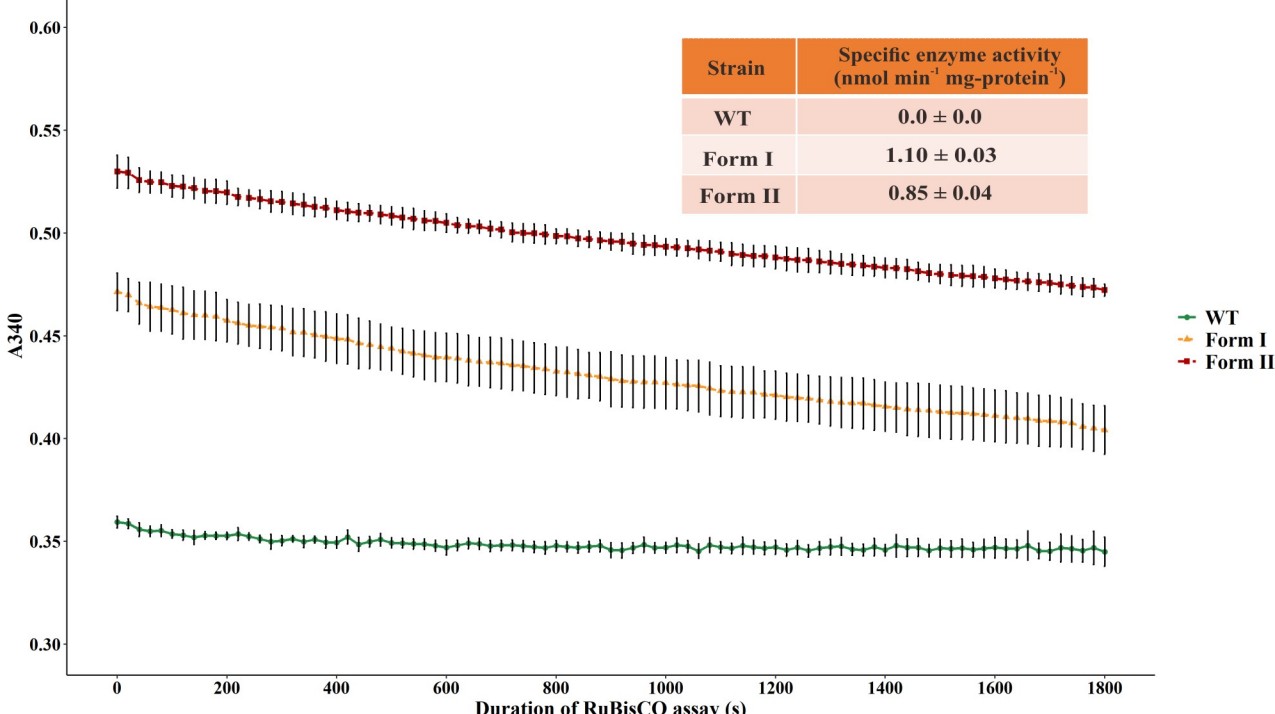

| Strain | Specific enzyme activity (nmol min$^{-1}$ mg-protein$^{-1}$) |
|---|---|
| WT | 0.0 ± 0.0 |
| Form I | 1.10 ± 0.03 |
| Form II | 0.85 ± 0.04 |

**Fig 3. Specific RuBisCO enzyme activity.** The line plot represents the decrease in $A_{340}$ absorbance, indicating NADH oxidation throughout the RuBisCO assay. The table displays specific RuBisCO activity of form I and form II RuBisCO.

in the Fig 4E. However, there was no statistically significant difference in ethanol concentrations produced by WT and RuBisCO strains ($p > 0.05$). It was likely that in the excess of carbon source condition, RuBisCO genes only played a negligible role in recombinant *K. marxianus* strains. This finding is in agreement with Joshi *et al.* [17] who found that form I RuBisCO is the most common form of RuBisCO under $CO_2$ limiting condition and responsible for providing cellular carbons. They hypothesized that this was a response of microorganisms to carbon limitation and may be important for scavenging the low levels of dissolved $CO_2$ to maintain growth and $CO_2$ fixation. In contrast, in YPD-0 when the growth of *K. marxianus* only based on amino acids and perhaps a trace amount of sugar, the engineered RuBisCO yeast strains grew faster and produced significantly higher ethanol concentration compared to the WT (Fig 4A and 4B). In Fig 4D, ethanol reached the highest concentration at 48 h and then gradually decreased in harmony with sugar depletion. The decrease in ethanol concentrations after 48 h could be explained by the reversible characteristics of alcohol dehydrogenase enzyme that could catalyze the conversion of ethanol back into acetaldehyde, especially when sugar depletes in the medium. In a previous study, two alcohol dehydrogenase enzymes KmAdh3 and KmAdh4 from *K. marxianus* GX-UN120, which were heterologously expressed in *E. coli*, exhibited the ability to catalyze the oxidation reaction of ethanol to acetaldehyde but not the reduction reaction of acetaldehyde to ethanol [31]. In addition, in a growth medium where ethanol was used as the sole carbon source, cell activates gluconeogenesis genes to shift to non-fermentative growth on $C_2$ and $C_3$ carbon sources. In *S. cerevisiae*, *ScGPM1* gene encoding phosphoglycerate mutase 1 showed the highest expression in this condition [32]. As phosphoglycerate mutase 1 gene *KmGPM1* (accession number KLMA_20098) in *K. marxianus* is homologous to *ScGPM1* with 87% identity, the use of ethanol to generate ATP and sugar phosphates for nucleotide biosynthesis, cell wall construction and storage carbohydrate

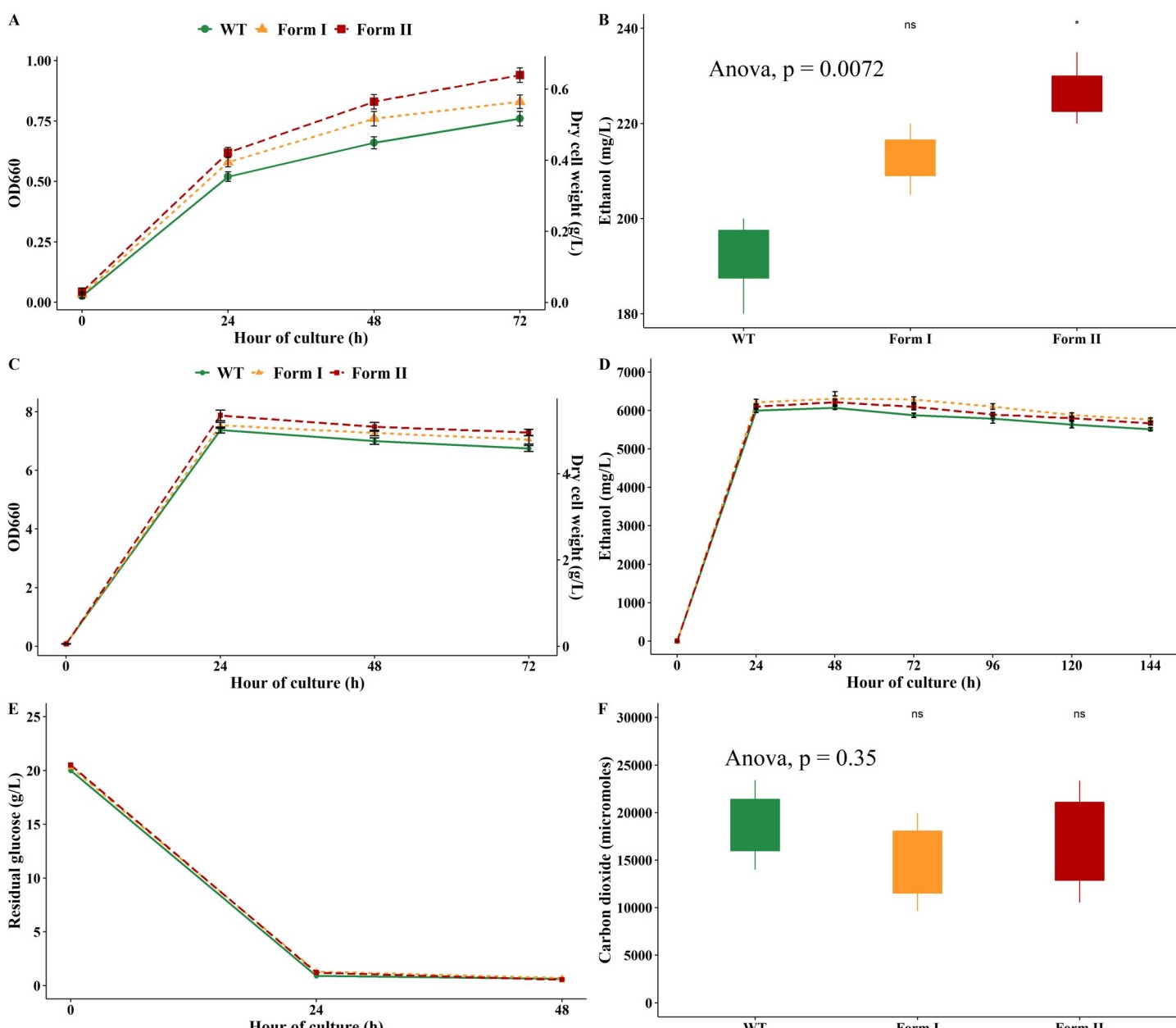

**Fig 4.** (A) Growth profiles of WT and RuBisCO *K. marxianus* strains in YPD-0 medium. (B) Ethanol concentrations of WT and RuBisCO *K. marxianus* strains in YPD-0 medium. (C) Growth profiles of WT and recombinant *K. marxianus* strains in YPD-20 medium. (D) Ethanol produced by WT and RuBisCO *K. marxianus* strains in YPD-20 medium. (E) Residual reducing glucose at the end of the fermentation in YPD-20 medium. (F) Accumulated carbon dioxide during the fermentation in YPD-20 medium.

biosynthesis should be taken into consideration under carbon depletion. The finding also suggested that with the ethanol fermentation conditions described above, ethanol distillation should not be carried out later than 48 hours of fermentation as its concentration is the highest at 48 h and might decrease soon. This is of importance in industrial ethanol production as it not only saves time but also reduces the operating cost. Regarding $CO_2$ production during the fermentation, it may be expected that $CO_2$-fixing enzymes function well to incorporate more $CO_2$ into the central carbon metabolism of the engineered strains. Although $CO_2$ produced by

WT was slightly greater than those of form I RuBisCO and form II RuBisCO strains, there was no statistically significant difference between 3 strains ($p > 0.05$). Specifically, in YPD-20 medium, the WT produced $0.44 \pm 0.094$ mmol $CO_2$/mmol glucose, followed by form II RuBisCO with $0.4 \pm 0.138$ mmol $CO_2$/mmol glucose and $CO_2$ produced by form I RuBisCO strain was the least with $0.35 \pm 0.11$ mmol of $CO_2$/mmol of glucose consumed. The results confirm the role of form I RuBisCO gene in $CO_2$ fixation and in agreement with previous reports as engineered RuBisCO *E. coli* [33] and RuBisCO *S. cerevisiae* [34] harboring RuBisCO genes released less $CO_2$ compared to the control.

### *K. marxianus* cultured in plant biomass hydrolysates

The WT, form I and form II RuBisCO *K. marxianus* strains were cultured in Napier grass and rice straw hydrolysates that were collected from the fermentation broths of *H. thermocellum* ATCC 27405 grown on Napier grass or rice straw powder. These hydrolysates were the end-products of the solubilization and bioconversion of plant biomass using the cellulolytic bacterium. (**Note:** The term hydrolysate and fermentation broth, therefore, were used interchangeably in this study). The schematic diagram presented in Fig 5 illustrated the general information regarding biomass powder preparation, co-culture *H. thermocellum-K. marxianus* system, and ethanol fermentation using biomass hydrolysates as substrates.

The ethanol, reducing sugar, and protein concentrations in the hydrolysate were determined as described in Materials and Methods section. At the end of the 6 days of fermentation by *H. thermocellum*, the collected Napier grass fermentation broth contained 940 mg/L ethanol, 8.1 g/L reducing sugar, and 10.3 mg/L protein. Similarly, the rice straw fermentation broth contained 970 mg/L ethanol, 7.4 g/L reducing sugar, and 8.8 mg/L protein. The reducing sugar yield in our Napier grass fermentation broth was similar to the value observed in the study of Amnuaycheewa *et al.* [23] (8.1 g/L vs. 7.66 g/L) in which pretreated Napier grass powder was used as a starting material.

On Napier grass and rice straw broth (with comparable amounts of reducing sugar and protein in the broth), the WT and engineered RuBisCO *K. marxianus* strains exhibited similar growth patterns (Fig 6A). In addition to the ethanol produced by *H. thermocellum* (940 mg/L in Napier grass hydrolysate and 970 mg/L in rice straw hydrolysate), the WT produced 670 mg/L and 710 mg/L ethanol, accounting for 41.6% and 42.1% increase in the final ethanol concentrations in Napier grass broth and rice straw broth, respectively. In addition, the ethanol yields achieved from RuBisCO strains were significantly greater than those from the WT ($p < 0.01$). In Napier grass broth, the additional amounts of ethanol produced by form I and form II RuBisCO strains was 880 mg/L and 850 mg/L, contributing 48.4% and 47.6% increase in the total ethanol production, respectively (Fig 6B).

In rice straw broth, similar pattern could be observed as form I and form II RuBisCO strains generated 780 mg/L and 860 mg/L extra ethanol, accounting for 44.4% and 47.0% increase to the final ethanol yields (Fig 6C). Although the starting soluble reducing sugars in both plant biomass broths are comparable, the composition of sugars might be different, resulting in slight differences in final ethanol yields. In Napier grass broth, form I RuBisCO strain produced approximately 11.2% higher ethanol than the WT did, and form II RuBisCO strain generated about 9.9% higher ethanol compared to the WT. In rice straw broth, the contribution of RuBisCO strains in enhancing ethanol yield was not as well as they did in Napier grass broth. The transformant form I and form II RuBisCO strains only produced 4.0% and 8.3% more ethanol relative to the WT, respectively (Fig 6C). In general, RuBisCO genes allowed the improvement of ethanol yields, suggesting their roles in $CO_2$ fixation and/or redox balance might contribute to ethanol production in the yeast cells during fermentation. In the

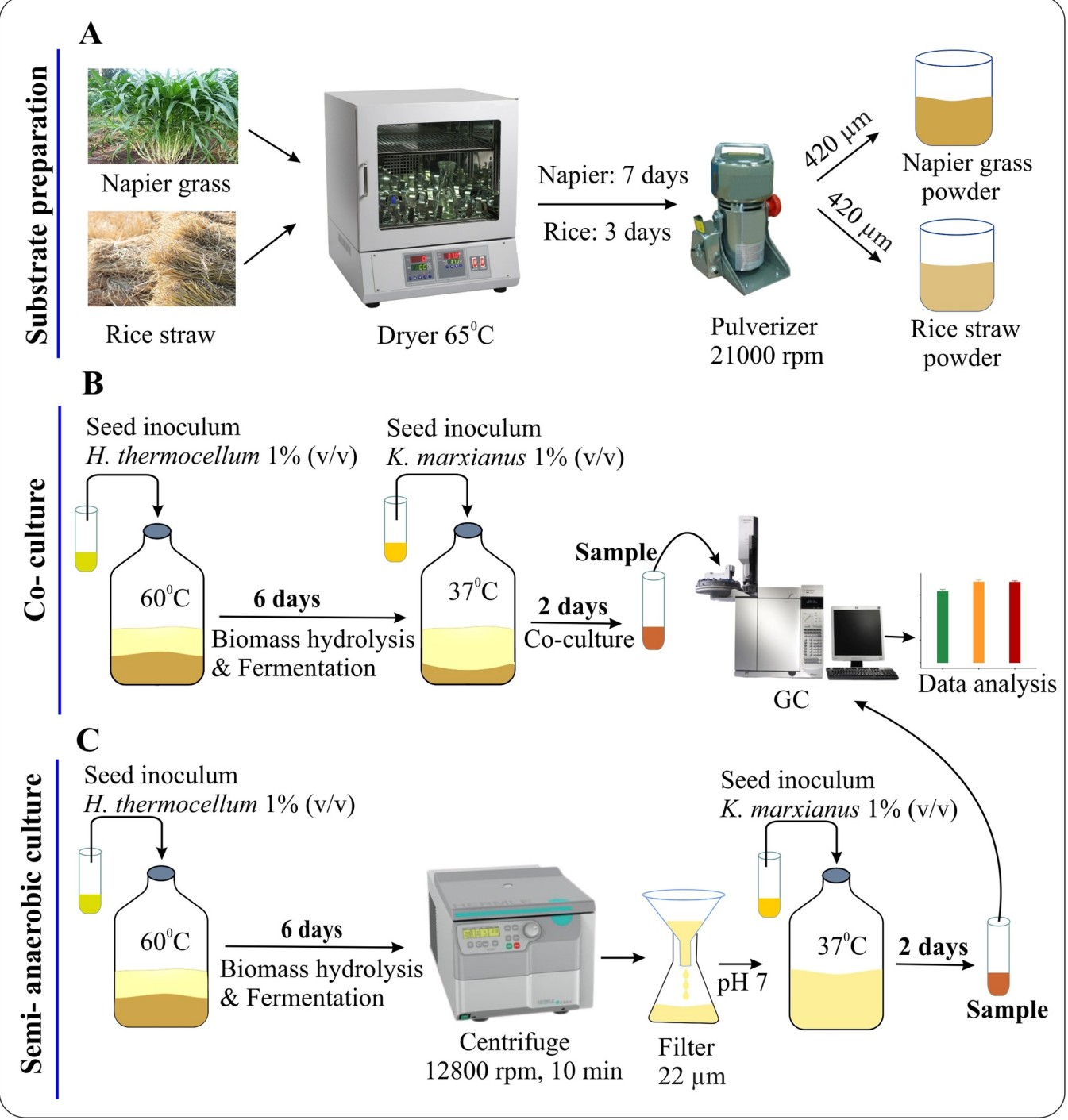

**Fig 5.** Schematic diagram of the present study (A) The procedure of substrate preparation (from field to powder). (B) The coculturing of *H. thermocellum* and *K. marxianus*. (C) The semi-anaerobic culturing of *K. marxianus* in Napier grass hydrolysate or rice straw hydrolysate.

YPD-8 where glucose, the favored carbon source of *K. marxianus*, was used, much faster growth patterns of all *K. marxianus* strains were observed compared to those in biomass hydrolysates (Fig 6D). In addition, the amount of bioethanol produced by the engineered RuBisCO strains was statistically greater than that produced by the WT.

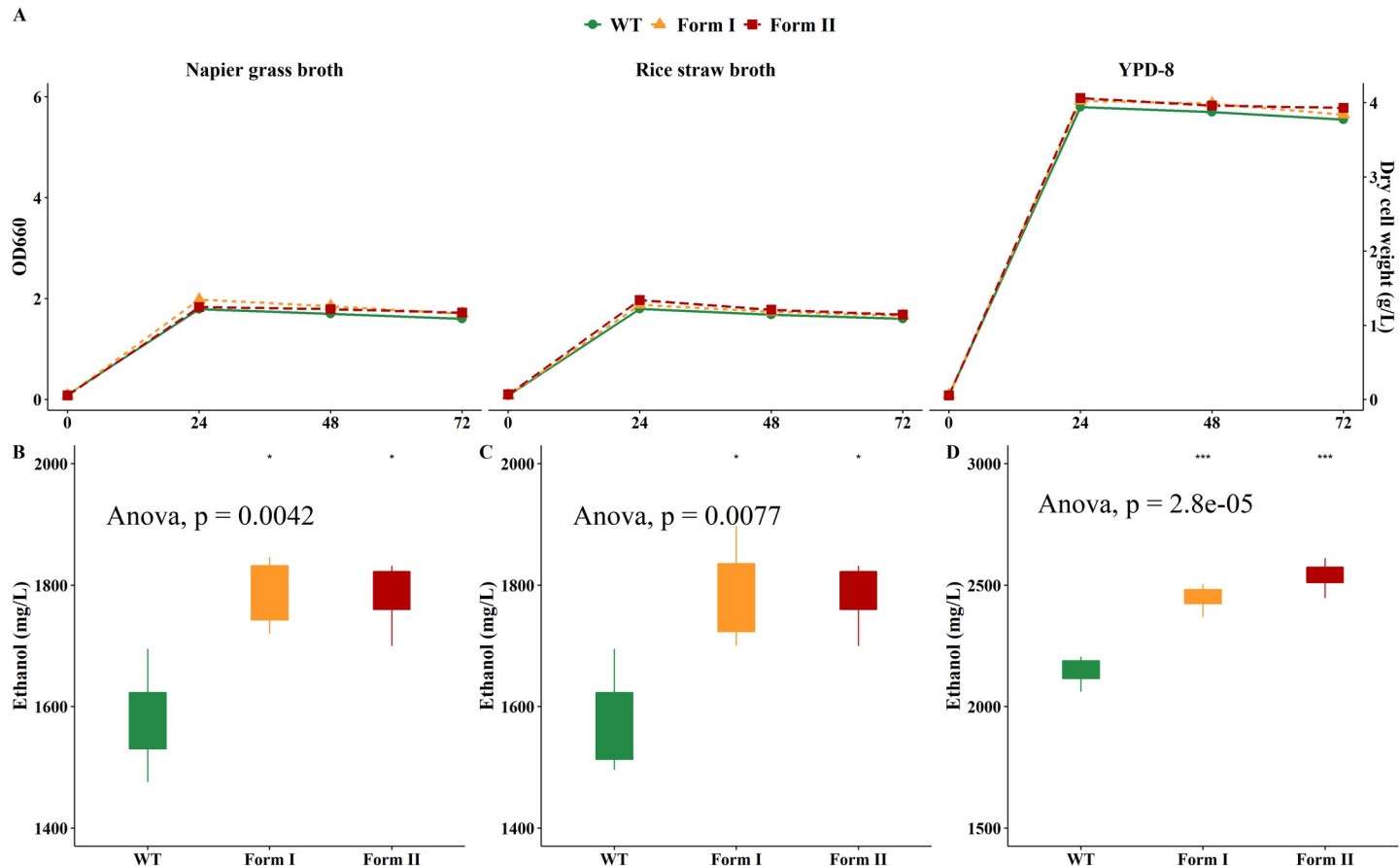

**Fig 6.** (A) Growth profiles of *K. marxianus* strains on Napier grass hydrolysate, rice straw hydrolysate and YPD-8. (B) Ethanol production of the WT, form I and form II RuBisCO *K. marxianus* strains grown on Napier grass hydrolysate. (C) Ethanol production of WT, form I and form II RuBisCO *K. marxianus* strains grown on rice straw hydrolysate. (D) Ethanol production of WT, form I, form II RuBisCO strains grown on YPD-8 medium.

To date, as effective consolidated bioprocessing (CBP) organisms remain to be developed, the two-step ethanol fermentation in the present study demonstrated its advantages. In the separate hydrolysis and fermentation (SHF) process, appropriate temperatures for hydrolytic enzymes and fermentation could be optimized independently. Moreover, as the hydrolysate solution could be sterilized, the risk of contamination could be reduced. The SHF process, however, requires longer time and the capital cost is higher than that for the simultaneous saccharification and fermentation (SSF) process [35].

## Efficiency in the utilization of complex sugars

After the fermentation process and the removal of the alcohol-producing microorganism, the recovered liquid composes water, ethanol, residual sugars, proteins and small amounts of organic acids [36]. In addition, a potent ability to take up and metabolize a wide range of sugars is especially important in an industrial process using plant biomass as the starting materials. However, in this study, a large amount of reducing sugars remained in the Napier grass broths (Fig 7).

Similar pattern was observed in the rice straw broth. This finding demonstrates that the efficiency of complex sugar utilization was not impressive and needs to be improved for a better ethanol production. The low capability in sugar consumption could be explained by the lack of

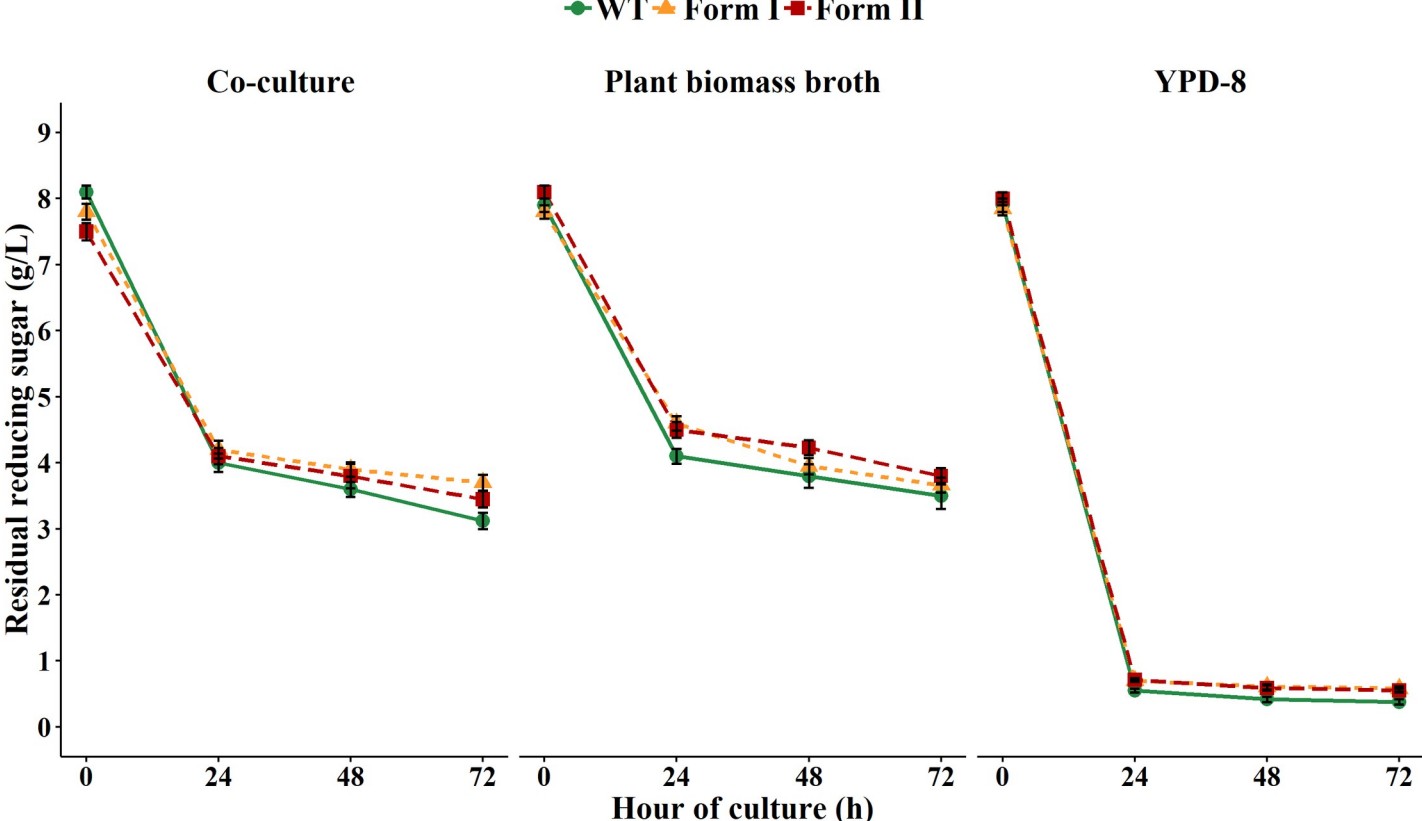

**Fig 7. Residual reducing sugar concentration profiles in the fermentation process.** The left panel shows the reducing sugars in the co-culture system (*H. thermocellum* + *K. marxianus*) in modified GS-2 medium supplemented with 100g/L Napier grass. The middle panel shows the residual reducing sugars in the Napier grass broth. The right panel shows the residual glucose in YPD-8 medium.

a robust sugar transport system and/or by the physiological/metabolic inhibition caused by toxic chemicals generated during the solubilization of biomass substrates [37]. The transformation of robust exogenous cellodextrin transporters, therefore, is of importance to facilitate the use of complex sugars in biomass broths as proved in previous studies [38,39]. In contrast, in the YPD-8 medium that was supplied with purified glucose (8 g/L), all *K. marxianus* strains grew much faster and metabolized sugar much more effectively than those grown in plant biomass broths. After 24 h of culture, most of glucose in the YPD-8 medium was consumed as shown in the Fig 7 (right panel). Consequently, the biomass of all yeast strains and ethanol yield in YPD-8 medium were much greater than those grown in the biomass hydrolysates. Regarding $CO_2$ production during the fermentation, it may be expected that $CO_2$-fixing enzymes function well to incorporate more $CO_2$ into the central carbon metabolism of the engineered strains. Although $CO_2$ produced by WT was slightly greater than those of form I RuBisCO and form II RuBisCO strains, there was no statistically significant difference between 3 strains ($p > 0.05$). Specifically, in YPD-8 medium, the WT produced $0.44 \pm 0.094$ mmol $CO_{2/}$ mmol glucose, followed by form II RuBisCO with $0.4 \pm 0.138$ mmol $CO_2$/mmol glucose and $CO_2$ produced by form I RuBisCO strain was the least with $0.35 \pm 0.11$ mmol of $CO_2$/ mmol of glucose consumed. Similar trend in the decrease of $CO_2$ emission was observed in RuBisCO strains grown in biomass hydrolysates, i.e., little lower than $CO_2$ generated by the WT but not statistically significant different ($p > 0.05$). The results confirm the role of form I RuBisCO gene in $CO_2$ fixation and in agreement with previous reports as engineered RuBisCO

*E. coli* [33] and RuBisCO *S. cerevisiae* [32] harboring RuBisCO genes released less $CO_2$ compared to the control.

## Co-culturing *K. marxianus* and *H. thermocellum* using Napier grass and rice straw powders as the substrates

Overall, the amount of ethanol generated by the co-culture fermentation system using *K. marxianus* and *H. thermocellum* was lower than that in the biomass broths. Specifically, the WT strain growth on Napier grass and rice straw powders generated 235 and 412 mg/L extra ethanol, contributing 20.0% and 29.8% to the final ethanol concentrations, respectively (Fig 8). On Napier grass substrate, recombinant form I and form II RuBisCO strains contributed 558 mg/L (37.3%) and 614 (39.6%) mg/L extra ethanol to the final ethanol concentrations, respectively (**Note:** The numbers in parentheses represent the percentage of contribution to the final ethanol concentrations). Similarly, on rice straw substrate, form I RuBisCO *K. marxianus* produced 496 mg/L (33.8%) and form II RuBisCO generated 576 mg/L (37.2%). In general, the RuBisCO strains produced ethanol better than the WT with 4.02–7.44% higher on rice straw and 17.3–19.5% higher on Napier grass.

The lower ethanol yields in co-culture system may be due to the nutrient competitions between ethanol producer and plant biomass degrader although *H. thermocellum* entered the stationary phase (after 144 h of culture) prior to the inoculation of *K. marxianus* into the co-culture vials. Moreover, low pH value of *H. thermocellum* culture broth (pH ~ 5.7), caused by the production of acetic acids via metabolic pathway of *H. thermocellum* itself and from the deacetylation of hemicellulose [40], might have inhibitory effects on kefir yeast growth and its metabolic functions. The plant biomass broths, in contrast, were thoroughly filtered to remove the bacterium *H. thermocellum* after the process of biomass solubilization. Furthermore, to optimize *K. marxianus* growth conditions, the broths were neutralized to pH 7.0. However, despite the suboptimal conditions for ethanol producer, co-culture system exhibits many advantages over monoculture by saving time and chemicals for culture broth collecting, filtering and neutralizing. It should be reminded that we only used unpretreated Napier grass and

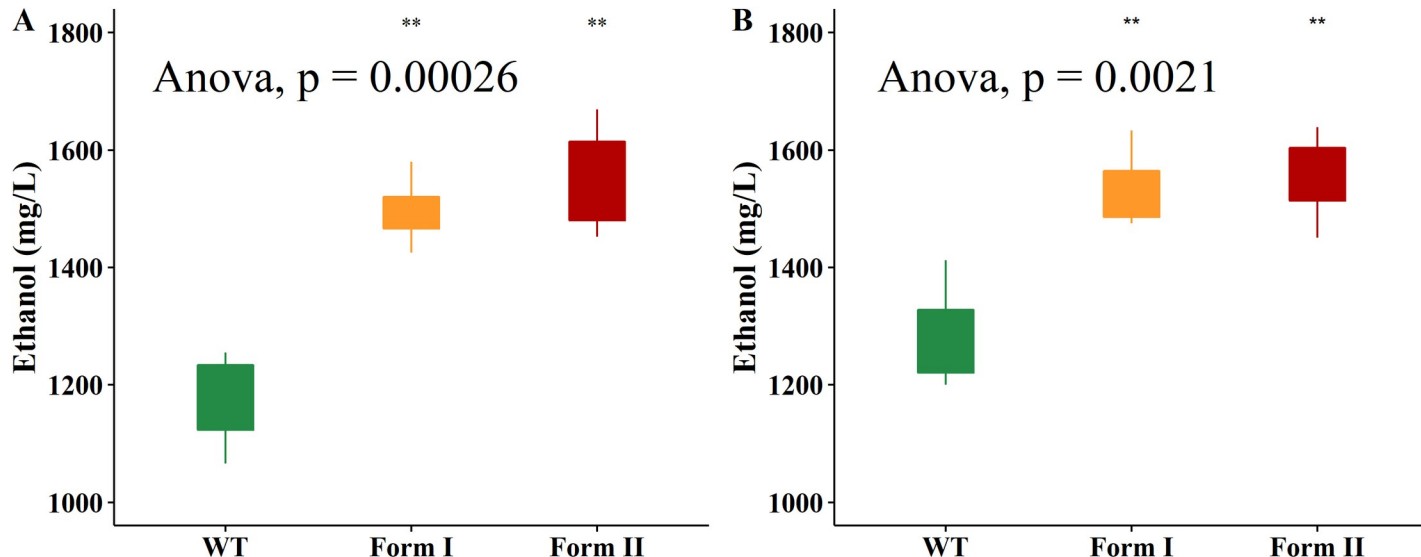

**Fig 8. Ethanol production in the co-culture *H. thermocellum-K. marxianus* system.** (A) Ethanol production with Napier grass powder as the substrate. (B) Ethanol production with rice straw powder as the substrate.

rice straw powder as starting material for bioconversion process to achieve similar results obtained in the study of Amnuaycheewa *et al.* [23] (8.08 g/L vs 7.66 g/L). These authors used pretreated Napier grass as a starting material, however, the hydrolysis step was performed using commercial cellulase enzymes (e.g., Celluclast (R) and Novozyme 188) instead of using a robust cellulolytic bacterium as ours. Our approach reduces the investment for facilities and environmental burden caused by chemicals used in the pretreatment process. However, it may negatively affect the conversion rate of plant biomass, resulting in low concentration of fermentable sugars released into the medium. In addition, toxic chemicals generated from biomass pretreatment, hydrolysis, and end-product formation may also negatively affect the growth and physiology of microorganism of interest for consolidated bioprocessing (CBP) [41–43].

## Conclusions

An economical cellulosic biofuel production system should meet three criteria, namely being an efficient process, a good biofuel producer and a good cellulolytic enzyme system [38]. In the present study, the use of a robust biomass-degrading bacterium for Napier grass and rice straw solubilization was proved efficient as it took advantage of hydrolytic machinery in *H. thermocellum* to decompose recalcitrant unpretreated plant biomass powders. The bioconversion of plant biomass not only released fermentable sugars for *K. marxianus*, but also produced more bioethanol, thus remarkably contributing to the final ethanol concentration. When growing in plant biomass hydrolysates, *K. marxianus* could use fermentable sugars left after the removal of *H. thermocellum* to synthesize 44–48% additional ethanol. In semi-anaerobic co-culturing fermentation system, may be due to the nutrient competition and/or suboptimal growth conditions, the contribution of *K. marxianus* to the final ethanol concentrations was approximately 20–39%. In terms of carbon utilization, except in YPD-20 medium, the RuBisCO yeast strains always exhibited better ethanol productivity than the WT, suggesting the important roles of RuBisCO genes in $CO_2$ fixation and/or in redox balancing.

## Supporting information

**S1 Raw images.**
(PDF)

## Acknowledgments

The authors would like to deeply acknowledge Dr. Nhuan Nghiem, Adjunct Professor, Biosystems Engineering Program, Department of Environmental Engineering and Earth Sciences, Clemson University, South Carolina 29636, USA for his careful English editing and valuable comments.

## Author Contributions

**Conceptualization:** Dung Minh Ha-Tran, Chieh-Chen Huang.

**Data curation:** Dung Minh Ha-Tran, Rou-Yin Lai, Eugene Huang.

**Formal analysis:** Dung Minh Ha-Tran, Trinh Thi My Nguyen.

**Funding acquisition:** Shou-Chen Lo, Chieh-Chen Huang.

**Investigation:** Eugene Huang, Shou-Chen Lo.

**Resources:** Chieh-Chen Huang.

**Supervision:** Chieh-Chen Huang.

**Validation:** Trinh Thi My Nguyen.

**Visualization:** Trinh Thi My Nguyen.

**Writing – original draft:** Dung Minh Ha-Tran, Chieh-Chen Huang.

**Writing – review & editing:** Eugene Huang, Shou-Chen Lo.

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
