## [Decision Letter · Decision Letter 0]

30 Nov 2020

PONE-D-20-34570

Construction of engineered RuBisCO Kluyveromyces marxianus for a dual microbial bioethanol production system

PLOS ONE

Dear Dr. Chieh-Chen Huang

Thank you for submitting your manuscript to PLOS ONE. After careful consideration, we feel that it has merit but does not fully meet PLOS ONE’s publication criteria as it currently stands. Therefore, we invite you to submit a revised version of the manuscript that addresses the points raised during the review process.

We look forward to receiving your revised manuscript.

Kind regards,

Muhammad Aamer Mehmood, Ph.D.

Academic Editor

PLOS ONE

Journal Requirements:

Reviewers' comments:

Reviewer's Responses to Questions

**Comments to the Author**

1. Is the manuscript technically sound, and do the data support the conclusions?

Reviewer #1: Partly

Reviewer #2: Yes

2. Has the statistical analysis been performed appropriately and rigorously? 

Reviewer #1: No

Reviewer #2: Yes

3. Have the authors made all data underlying the findings in their manuscript fully available?

Reviewer #1: No

Reviewer #2: Yes

4. Is the manuscript presented in an intelligible fashion and written in standard English?

Reviewer #1: Yes

Reviewer #2: Yes

5. Review Comments to the Author

Reviewer #1: Authors have expressed two variants of Rubiscos (Form I and Form II) to enhance the CO2 fixation and ultimately the ethanol production using these engineered strains. Authors report an extensive metabolic engineering of these pathways in a yeast K. marxianus strain KY3 in a hope to make the strain more efficient under CO2-limiting growth conditions. There are around seven genes/cassettes used for each form of the rubisco present in Rhodopseudomonas palustris. The authors do not show: a) that the cassettes were actually integrated into the yeast genome, b) each of the cassette/gene was expressed in the cells, c) whether the functional forms of Rubisco were actually assembled, and d) activity of recombinant rubiscos in CO2-limiting as well as non-limiting CO2 conditions . At present the results are based on the assumptions that recombinant Rubsicos were successfully assembled and that they were in a biochemically active conformation. Unless, this part of the paper is addressed, the paper cannot be accepted in its present form.

Reviewer #2: In this article, K. marxianus was used as a host for the transformation of form I and form II RubisCO genes derived from the nonsulfur purple bacterium R. palustris. H. thermocellum was used to degrade Napier grass and rice straw to generate soluble fermentable sugars. The RuBisCO expression of two engineered K. marxianus was analyzed. Growth profiles, ethanol production, and residual reducing glucose/carbon dioxide were also detected and compared in different conditions about WT and engineered strains. This research is very close to industrial applications, but this study is still relatively primary. I’d like to encourage the authors to address the following concerns.

Major：

1. Please clarify the definitions of form I and form II RubisCO genes in the Introduction.

2. In Materials and Methods, how to efficiently clone amplified gene fragments into the designated cassettes? And how to insert multiple exogenous genes into K. marxianus genome. Please give details.

3. In Figure 2B, there are doubts about the corresponding relationship between the electrophoresis bands and the sizes. Eg: left 4 and right 2.

4. Please explain the reason why the expression is lower than 1.0 in Figure 3B.

5. To make the article more logical and clearer, the section “Growth Profiles and Ethanol ….Medium YPD-20” should not be arranged at the end of this manuscript.

Minor:

1. Line 103, there is a lack of punctuation at the end of the sentence.

2. Line 162, “FeSO4.7H20” should be changed to “FeSO4·7H2O”. Line 190 is the same.

3. Line 121, “ antibiotics” should be changed to “antibiotic”.

4. Line 180 and 186, unify the speed unit (rpm/x g). “12.800” should be changed to “12,800”. Please handle similar errors as well.

5. The OD of H. thermocellum is far greater than K. marxianus, can changing temperature overcome growth advantages (60℃-37℃)?

6. “After 144 h, the culture stopped producing H2 and CO2, indicating the end of metabolic activities.” The measurement of hydrogen is not mentioned in this paper. Please explain how to know the H2 production ceased.

7. The size of letters should be constant in Figures 4 and 7. Especially for Figure 7, some words are too small to read.

8. Ethanol concentration should use “g/L” as a unit.

9. Is there any special reason to choose Box plot for ethanol production? Why not use a line plot like residue sugar. Actually, sugar and ethanol can be put into one figure.

6. PLOS authors have the option to publish the peer review history of their article (what does this mean?). If published, this will include your full peer review and any attached files.

Reviewer #1: **Yes: **Niaz Ahmad

Reviewer #2: No

---

## [Author Response · Author response to Decision Letter 0]

29 Jan 2021

Dear Reviewer 1

We appreciate your constructive comments on our work. We have carefully considered the comments and have addressed points by points in the reply. Furthermore, some parts of the manuscript have been added or modified in response to your suggestion and comments.

Reviewer #1: Authors have expressed two variants of Rubiscos (Form I and Form II) to enhance the CO2 fixation and ultimately the ethanol production using these engineered strains. Authors report an extensive metabolic engineering of these pathways in a yeast K. marxianus strain KY3 in a hope to make the strain more efficient under CO2-limiting growth conditions. There are around seven genes/cassettes used for each form of the rubisco present in Rhodopseudomonas palustris. The authors do not show: 

a) That the cassettes were actually integrated into the yeast genome

Response: The results showed in the Figure 1B and the Figure 1D were PCR products of the combined gene cassettes that were amplified from the engineered RuBisCO yeast genome (e.g., cassette 1 + cassette 2, cassette 2 + cassette 3, etc.). The gel images confirmed that the RuBisCO gene cassettes were integrated into the host genome in a predesignated order as described in the manuscript and were illustrated in the Figure 1A, 1C. Furthermore, the RT-qPCR data showed in the Figure 2 revealed that the RuBisCO genes were appropriately inserted into the K. marxianus genome and functioned well. We already revised and corrected this in the revised manuscript.

b) Each of the cassette/gene was expressed in the cells

Response: In this study, we confirmed the genes integration into the host genome and the transcription levels of these RuBisCO genes as mentioned above. Moreover, the RuBisCO activities were confirmed, and the phenotypes such as G418 resistance trait and the elevated ethanol production of the engineered RuBisCO strains compared to the WT strain help to confirm the presence of transgenes and their expected functions in the host genome. 

c) Whether the functional forms of Rubisco were actually assembled?

Response: Although we did not purify the assembled Rubisco to confirm the functional form of RuBisCO from the host cells, we demonstrated the RuBisCO enzymatic activities as described in the revised manuscript. 

d) Activity of recombinant RuBisCO in CO2-limiting as well as non-limiting CO2 conditions. At present the results are based on the assumptions that recombinant RuBisCO were successfully assembled and that they were in a biochemically active conformation. 

Response: As the culture media contained organic carbon sources, and the CO2 was generated from metabolic activities of microorganism, it is hard to determine whether the condition was CO2-limiting or CO2 non-limiting. However, the activities of RuBisCO were confirmed. Here is the added paragraph to the Materials and Methods. This paragraph was highlighted in the revised manuscript using Track Changes. The assay showed evidence that the RuBisCO genes function well in the K. marxianus cells and the RuBisCO enzyme was synthesized. 

Activity assay of RubisCO

RuBisCO activity measurement was performed based on the continuous spectrophotometric rate determination method described in [14,27] with some modifications. The crude cell lysate was prepared using the glass bead lysis as described by Mukherjee et al. [28]. Briefly, the yeast strains were grown semi-anaerobically in 250-mL serum bottles containing YPD-8 medium for 12 h. Cells were harvested by centrifuging at 12,800 x g, 10 min at 4oC, washed twice with sterile distilled water, and then resuspended in 350 µL of lysis buffer containing 50 mM Tris-HCl buffer, pH 7.5, with 2 mM EDTA, 0.1 mM phenylmethylsulfonyl fluoride (PMSF), 0.1 mM DTT, 3 mM MgCl2, 10 mM NaCl. The cells in the glass bead tube were vortexed in a cyclomixer for 4 times at 2,500 rpm for 15 s each, immersing the cell suspension in ice for 15 s between 2 vortexing cycles. The cell suspension was centrifuged at 5,000 x g for 10 min at 4oC to collect the supernatant and the supernatant was used as the crude cell lysate for enzymatic assay. The reaction mixture containing cell lysate, 50 mM HEPES buffer (pH 8), 1 mM Ribulose 1,5-bisphosphate (RuBP), 20 mM MgCl2, 5 mM DTT, 20 mM NaHCO3, 5 mM ATP, 0.15 mM NADH, 5 U/mL 3-phosphoglycerate kinase, 6 U Glyceraldehyde 3-phosphate dehydrogenase was prepared. Specifically, 20 µL cell lysate was added to 180 µL of reaction mixture. To initiate the reaction, 7 µL of 33 mM RuBP was added to the reaction mixture to reach a final concentration of 1 mM. The absorbance at 340 nm (A340) was then recorded every 20 s for 30 min using Beckman Coulter - PARADIGM™ microplate reader (Beckman Coulter, USA) and the rate of decrease in A340 was then converted to the rate of NADH oxidation. RuBisCO activity was calculated from the rate of NADH oxidation. The millimolar extinction coefficient of NADH at 340 nm was used to determine the rate of NAD+ production. Total protein concentration in the cell-free extracts was measured by the Bradford protein assay (Bio-Rad Protein Assay).

Formula for RuBisCO enzyme activity calculation:

ΔA340: Initial A340 – Final A340

2: 2 µmoles of β-NADH are oxidized for each µmole of D-Ribulose 1,5-diphosphate used.

6.22: Millimolar extinction coefficient (mmol-1 cm-1) for β-NADH at 340 nm.

0.6: Optical path length (cm) in an ELISA well with 200 μL reaction volume.

Dear Reviewer 1

We hope that our responses and correction are satisfactory to you, and that it is now suitable for publication in PLOS ONE Journal.

Dr. Chieh-Chen Huang

Dear Reviewer 2

First of all, we would like to appreciate your constructive comments and suggestions on our work. We have carefully considered the comments and have clearly addressed in the reply. Parts of the manuscript have been modified in response to your suggestion and comments.

Reviewer #2: In this article, K. marxianus was used as a host for the transformation of form I and form II RubisCO genes derived from the nonsulfur purple bacterium R. palustris. H. thermocellum was used to degrade Napier grass and rice straw to generate soluble fermentable sugars. The RuBisCO expression of two engineered K. marxianus was analyzed. Growth profiles, ethanol production, and residual reducing glucose/carbon dioxide were also detected and compared in different conditions about WT and engineered strains. This research is very close to industrial applications, but this study is still relatively primary. I’d like to encourage the authors to address the following concerns.

Major：

1. Please clarify the definitions of form I and form II RubisCO genes in the Introduction.

Response: We added a new paragraph to the Introduction to clarify the definitions of form I and form II RubisCO cassettes as your request. The added paragraph was highlighted using the “Track Changes” function in Microsoft Word. In addition, the Figure 1A and Figure 1C illustrated the form I and form II RubisCO cassette constructions as described in the PGASO method.

“The genome of R. palustris consists of two active forms of RuBisCO. The form I RuBisCO includes cbbL (large subunit), cbbS (small subunit), cbbR (transcription regulator) and cbbRRS (atypical two-component systems) genes. The cbbL and cbbS genes are located at the distal end of the cbbI operon and the cbbRRS genes are found located between the cbbR and cbbLS genes. The form II RuBisCO comprises cbbM gene which is located at the distal end of the cbbII operon, followed by cbbA (fructose 1,6-bisphosphate aldolase), cbbT (transketolase), cbbP (phosphoribulokinase), and cbbF (fructose 1,6/sedoheptulose 1,7-bisphosphatase) genes. In the present study, the cbbL, cbbS, and cbbP genes were chosen to construct the form I RuBisCO cassette and the cbbM, cbbP genes were selected to construct the form II RuBisCO cassette [14,15]”

2. In Materials and Methods, how to efficiently clone amplified gene fragments into the designated cassettes? And how to insert multiple exogenous genes into K. marxianus genome. Please give details.

Response: The PGASO method used in this paper relies on the use of overlapping gene fragments and it could be applicable to any host that can undergo homologous recombination. Each gene cassette contains a specific promoter, gene fragment, and a terminator. In the first gene cassette, the 1529 bp sequence identical to the 3’ region sequence of K. lactis Lac4 promoter was used as a promoter, and at the 3’ end of the terminator (TTLac), a 55 bp sequence was designed to be homologous to the 5’ end of the promoter ScPGapDH in the gene cassette 2. The terminator ScTTGap of the gene cassette 2, in turn, had 55 bp overhanging sequence at its 3’ end that was homologous to the 5’ end of the promoter ScPGK in the gene cassette 3, and so on. We hope that the illustration in the Figure 1A and Figure 1C make the PGASO concept clearer to the readers.

In addition, the PGASO method in Materials and Methods was also rewritten to make it easier to understand. This paragraph was highlighted using Track Changes. 

“Multiple-gene cassette construction

 The PGASO method was developed to insert multiple exogenous genes into K. marxianus 4G5 genome [21]. Since the PGASO method was fundamentally based on site-specific homologous recombination, overhanging sequences were designed to link at the 5’-upstream sequence of a promoter, and to link at the 3’-downstream sequence of a terminator in order to facilitate an accurate gene cassette assembly into a host genome. Specifically, in the first gene cassette, a 1529 bp sequence, which is identical to the 3’ region of the K. lactis Lac4 promoter, was used as a promoter in the first gene cassette. In the last gene cassette, a 582 bp sequence, which is homologous to the 5’ region of K. lactis Lac4 promoter, was linked with the 3’ end of the terminator ScTTADHI. It is noteworthy that the identities between K. marxianus Lac4 promoter and K. lactis Lac4 promoter, at the 5’ region and at the 3’ region, are 99.8% and 97.9%, respectively. The other gene cassettes were constructed to contain two parts as follows: (1) a gene sequence, at its 5’ end, linked to a promoter, at its 3’ end, linked to a terminator, and (2) a 55 bp overhanging sequence at the 3’ end of the gene cassette that is homologous to a 5’ end of the promoter of its downstream neighboring gene cassette. Individual gene fragment of cbbL, cbbS, cbbM, and cbbP from the genome of R. palustris CGA009 were amplified by polymerase chain reaction (PCR). The amplified gene fragments cbbL, cbbS, cbbM, cbbP and G418 (kanamycin resistance gene) were then cloned into the predesignated cassette plasmids (Table 1) with their specific independent promoter and terminator as follows: ScPGapDH-cbbL¬-ScTTGap, KlPGapDH-G418-ScTTGap, KlPADHI-cbbS-ScTTGap, ScPADHI-cbbP-ScTTADHI, and KlPPGK-cbbM-ScTTPGK. Subsequently, gene cassettes for PGASO technique were amplified with KOD plus DNA polymerase kit (TOYOBO Biotech) with specific primers listed in the Table 1. The cloning procedure was performed using Escherichia coli strain DH5α cells and Luria-Bertani (LB) medium was used as a culture medium. The antibiotic ampicillin (50 μg/mL) was used to screen the cloned plasmids. The sizes of individual transgenes and gene cassettes were shown in the Table 2. In addition, specific primers for RT-qPCR were designed to confirm the transcriptional activities of the transgenes cbbS, cbbL, cbbM, and cbbP in the host cells. Importantly, to verify the sizes and the correct orders of the gene cassettes in their predesignated assemblage in the yeast genome, the combined gene cassettes were also amplified using PCR with the forward primer of the upstream gene cassette and the reverse primer of its adjacent downstream gene cassette (Table 1).

Table 1. Plasmids and primers used in the PGASO constructions and RT-qPCR experiments.

Plasmid Description

pUC18 Ampicillin resistant; multicopy plasmid with a ColE1 – type replicon

Cassette 1 pUC18-Kl PLac4 pUC18 derivative with a 1529 bp portion of Kluyveromyces lactis Lac4 promoter and K. lactis Lac4 terminator with a 55 bp sequence at 3’ end homologous to S. cerevisiae GapDH promoter

Cassette 2 pUC18-Sc PGapDH pUC18 derivative with S. cerevisiae GapDH promoter and TTGap terminator with a 55 bp sequence at 3’ end homologous to S. cerevisiae PGK promoter

Cassette 3 pUC18- Sc PPGK pUC18 derivative with S. cerevisiae PGK promoter and TTPGK terminator with a 55 bp sequence at 3’ end homologous to K. lactis GapDH promoter

Cassette 4 pUC18- Kl PGapDH pUC18 derivative with K. lactis GapDH promoter S. cerevisiae GapDH terminator with a 55 bp overhanging sequence at its 3’ region homologous to K. lactis PGK promoter

Cassette 5 pUC18-Kl PPGK pUC18 derivative with K. lactis PGK promoter and S. cerevisiae PGK terminator with a 55 bp sequence at 3’ end homologous to K. lactis ADHI promoter

Cassette 6 pUC18-Kl PADHI pUC18 derivative with K. lactis ADHI promoter and S. cerevisiae GapDH terminator with a 55 bp sequence at 3’ end homologous to S. cerevisiae ADHI promoter

Cassette 7 pUC18-Sc PADHI pUC18 derivative with S. cerevisiae ADHI promoter and ADHI terminator with a 582 bp sequence at 3’ end homologous to K. lactis Lac4 promoter

Cassette 2 pUC18-Sc PGapDH-cbbL Cassette 2 pUC18-Sc PGapDH derivative with cbbL

Cassette 6 pUC18-Kl PADHI-cbbS Cassette 6 pUC18-Kl PADHI derivative with cbbS

Cassette 5 pUC18-Kl PPGK-cbbM Cassette 5 pUC18-Kl PPGK derivative with cbbM

Cassette 7 pUC18-Sc PADHI-cbbP Cassette 7 pUC18-Sc PADHI derivative with cbbP

Cassette 4 pUC18- Kl PGapDH-G418 Cassette4 pUC18- KlPGapDH derivative with kanR from Lac4-KanMX cassette [21]

Cassette Primer Sequence

KlPLac4-KlTTLac4 KlPLac4-3’ 5’-CCGCGGGGATCGACTCATAAAATAG-3’

 KlTTLac4_ScPGapDH 5’-CTACTATTAATTATTTACGTATTCTTTGAAATG

GCAGTATTGATAATGATAAACTTATACAACATCG

AAGAAGAGTCT-3’

ScPGapDH-cbbL-ScTTGapDH ScPGapDH 5’-AGTTTATCATTATCAATACTGCCAT-3’

 ScTTGap_ScPGK 5’-GGACTCCAGCTTTTCCATTTGCCTTCGCGCTT

GCCTGTACGGTCGTTACCATACTTGGCGGAAAA

AATTCATTTGTAA-3’

ScPPGK-ScTTPGK ScPGK 5’- ACTGTAATTGCTTTTAGTTGTGTAT-3’

 ScTTPGK_KlPGapDH 5'-GGACTCCAGCTTTTCCATTTGCCTTCGCGCTT

GCCTGTACGGTCGTTACCATACTAAGCTTTTTCG

AAACGCAGAATTTTCG-3’

KlPGapDH-G418-ScTTGapDH Kl-PGapDH 5’-AGTATGGTAACGACCGTACAGGCAA-3’

 ScTTGap_KlPGK 5'-TACCTTTGATACCATAAAAACAAGCAAATATTCT

TACTTCAAACACACCCGTGGCGGAAAAAATTCATTTGTAAACT-3’

KlPPGK-cbbM-ScTTPGK KlPGK 5’-CGGGTGTGTTTGAAGTAAGAATATT -3’

 ScTTPGK_KlPADHI 5’-AGGTAAGTATGGTAACGACCGTACAGGCAAG

CGCGAAGGCAAATGGAAAAGCTGGAAGCTTTT

TCGAAACGCAGAATTTT-3’

KlPADHI-cbbS-ScTTGapDH KlPADHI 5’-CCAGCTTTTCCATTTGCCTTCGCGCTTGCC-3’

 ScTTGap_ScPADHI 5’-GGAATCCCGATGTATGGGTTTGGTTGCCAGA

AAAGAGGAAGTCCATATTGTACACTGGCGGAA

AAAATTCATTTGTAA-3’

ScPADH1-cbbP-ScTTADH1 ScPADHI 5’-GTGTACAATATGGACTTCCTCTTTTC-3’

 ScTTADHI_KlPLac4-5’End 5’-GAAATTTAGGAATTTTAAACTTG-3’

Primers used for cloning

Phosphorobulokinase (PRK) cbbP-F’ 5’-CCGCCTAGGATGTCCCGTAAGCATCCG-3’

 cbbP-R’ 5’-CATGCCATGGTTACTTCATGCTTCGTTTGCGGTC-3’

Form I RuBisCO small subunit cbbS-F’ 5’-CCGCCTAGGATGCGACTGACCCAAGGC-3’

 cbbS-R’ 5’-CATGCCATGGTCAGCCTCCGTAGCGTCG-3’

Form I RuBisCO large subunit cbbL-F’ 5’-CCGCCTAGGATGAACGAAGCAGTCACC-3’

 cbbL-R’ 5’-CATGCCATGGTTAGACCGAGACCGACGG-3’

Form II RuBisCO cbbM-F’ 5’-CCGCCTAGGATGGACCAGTCGAACC-3’

 cbbM-R’ 5’-CCTTAATTAATTACGCCGCCTGC-3’

Primers used for RT-qPCR

Phosphorobulokinase (PRK) r-cbbP-F’ 5’-GCCGATGAATCGATGGTGG-3’

 r-cbbP-R’ 5’-TTCATGCTTCGTTTGCGGTC-3’

Form I RuBisCO small subunit r-cbbS-F’ 5’-TGACCCAAGGCTGTTTCTCG-3’

 r-cbbS-R’ 5’-TTCAGTTCCATCATCACGCC-3’

Form I RuBisCO large subunit r-cbbL-F’ 5’-CCGGCGTGATGGAATACAAG-3’

 r-cbbL-R’ 5’-ATACTTCTCCGCCGCAGTCA-3’

Form II RuBisCO r-cbbM-F’ 5’-ATGATCGCCTCGTTCCTGAC-3’

 r-cbbM-R’ 5’-TTGATGATGGTGCCGACGAT-3’

Actin 4G5 actin F 5’-GGGCTTCGGTCAACAAAC-3’

 4G5 actin R 5’-TGGTCGGTATGGGTCAAAAGG-3’

Primers to confirm gene cassettes order in K. marxianus genome

Cassette (1 + 2 cbbL) 1-2 F 5’-CCGCGGGGATCGACTCATAAAATAG-3’

 1-2 R 5’-GGACTCCAGCTTTTCCATTTGCCTTCGCGCTT

GCCTGTACGGTCGTTACCATACTTGGCGGAAAA

AATTCATTTGTAA-3’

Cassette (2 cbbL + 3) 2-3 F 5’-AGTTTATCATTATCAATACTGCCAT-3’

 2-3 R 5'-GGACTCCAGCTTTTCCATTTGCCTTCGCGCTT

GCCTGTACGGTCGTTACCATACTAAGCTTTTTCG

AAACGCAGAATTTTCG-3’

Cassette (3 + 4 G418) 3-4 F 5’- ACTGTAATTGCTTTTAGTTGTGTAT-3’

 3-4 R 5'-TACCTTTGATACCATAAAAACAAGCAAATATTCT

TACTTCAAACACACCCGTGGCGGAAAAAATTCATTTGTAAACT-3’

Cassette (4 G418 + 5) 4-5 F 5’-AGTATGGTAACGACCGTACAGGCAA-3’

 4-5 R 5’-AGGTAAGTATGGTAACGACCGTACAGGCAAG

CGCGAAGGCAAATGGAAAAGCTGGAAGCTTTT

TCGAAACGCAGAATTTT-3’

Cassette (5 + 6 cbbS) 5-6 F 5’-CGGGTGTGTTTGAAGTAAGAATATT -3’

 5-6 R 5’-GGAATCCCGATGTATGGGTTTGGTTGCCAGA

AAAGAGGAAGTCCATATTGTACACTGGCGGAA

AAAATTCATTTGTAA-3’

Cassette (6 cbbS + 7 cbbP) 6-7 F 5’-CCAGCTTTTCCATTTGCCTTCGCGCTTGCC-3’

 6-7 R 5’-GAAATTTAGGAATTTTAAACTTG-3’

Table 2. Size of individual RuBisCO genes, gene cassettes used in the PGASO constructions.

Cassette Promoter-Terminator (bp) RubisCO gene (bp) Promoter-Gene-Terminator (bp) Primers

1 2347 - KlPLac4- KlTTLac4 (2347) F-KlPLac4-3’

R-KlTTLac4_ScPGapDH

2 1102 cbbL (1457) ScPGapDH-cbbL-ScTTGap (2559) F-ScPGapDH

R-ScTTGap_ScPGK

3 1294 - ScPPGK-ScTTPGK (1294) F-ScPGK

R-ScTTPGK_KlPGapDH

4 1145 G418 (810) KlPGapDH-G418-ScTTGap (1955) F-Kl-PGapDH

R-ScTTGap_KlPGK

5 1291 cbbM (1386) KlPPGK-cbbM-ScTTPGK (2677) F-KlPGK

R-ScTTPGK_KlPADHI

6 1301 cbbS (425) KlPADHI-cbbS-ScTTGap (1726) F-KlPADHI

R-ScTTGap_ScPADHI

7 2068 cbbP (876) ScPADHI-cbbP-ScTTADHI (2944) F-ScPADHI

R-ScTTADHI_KlPLac4-5’End

3. In Figure 2B, there are doubts about the corresponding relationship between the electrophoresis bands and the sizes. E.g.: left 4 and right 2.

Response: You are right. This was our mistake when using the wrong gel images. We already carefully revised them and displayed the correct gel images in the Figure 1B and Figure 1D.

4. Please explain the reason why the expression is lower than 1.0 in Figure 2.

Response: We used actin as the calibrator gene, thus the gene expression levels of RuBisCO were calculated relatively based on the Ct value of the actin gene. Although the expression level of cbbL was low, the RuBisCO activity, however, could be determined. We already checked the RT-qPCR results again and displayed the new results in the Figure 2.

5. To make the article more logical and clearer, the section “Growth Profiles and Ethanol ….Medium YPD-20” should not be arranged at the end of this manuscript.

Response: We already rearranged the section as your request. It is now placed between the RuBisCO assay and the K. marxianus cultured in plant biomass hydrolysates part.

Minor:

1. Line 103, there is a lack of punctuation at the end of the sentence.

Response: We already corrected it. 

2. Line 162, “FeSO4.7H20” should be changed to “FeSO4·7H2O”. Line 190 is the same.

Response: This is the typo, we already corrected it.

3. Line 121, “ antibiotics” should be changed to “antibiotic”.

Response: We already corrected this word as your request.

4. Line 180 and 186, unify the speed unit (rpm/x g). “12.800” should be changed to “12,800”. Please handle similar errors as well.

Response: We changed 12.800 to 12,800 based on your comment.

5. The OD of H. thermocellum is far greater than K. marxianus, can changing temperature overcome growth advantages (60℃-37℃)?

Response: We decreased the culture temperature from 60oC (for H. thermocellum) to 37oC for K. marxianus after H. thermocellum entered its decline phase. In other word, after 6 days of culture, H. thermocellum finished the biomass hydrolysis step and only maintained its low activity. Subsequently, the decrease in culture temperature to 37oC was suitable for the K. marxianus growth. By this way, we tried to prevent the potent competition between two species in the co-culturing system.

6. “After 144 h, the culture stopped producing H2 and CO2, indicating the end of metabolic activities.” The measurement of hydrogen is not mentioned in this paper. Please explain how to know the H2 production ceased.

Response: We measured H2 production at the same time with CO2 using the same GC protocol. We already modified this part to make it clearer.

7. The size of letters should be constant in Figures 4 and 7. Especially for Figure 7, some words are too small to read.

Response: We already redrew all the graphics with bigger font size as your request

8. Ethanol concentration should use “g/L” as a unit.

Response: We agree with the reviewer that ethanol concentration should be displayed as g/L, but we also think mg/L is acceptable. Therefore, we decided to use this unit consistently in this paper.

9. Is there any special reason to choose Box plot for ethanol production? Why not use a line plot like residue sugar. Actually, sugar and ethanol can be put into one figure.

Response: In the case of residual sugar, we measured sugar concentrations at different timepoints from the beginning of the culture (0 h), intermediate timepoints to the end of the process (24 h, 48 h, and 72 h) to observe the patterns of sugar consumption by K. marxianus. Therefore, line plot is a suitable way to describe the data. Ethanol measurement, in contrast, was only carried out at the end of the fermentation and the ANOVA was calculated based on the raw data collected from biological triplicates. In addition, boxplot is a useful way to show the contribution of data, their central values, and their variability. The use of boxplot is preferable to the barplot (with mean and standard deviation) when the data are not strictly symmetrically distributed. 

Dear Reviewer 2

We hope that our responses and correction are satisfactory to you, and that it is now suitable for publication in PLOS ONE Journal.

Dr. Chieh-Chen Huang

---

## [Editor Report · Decision Letter 1]

2 Feb 2021

Construction of engineered RuBisCO Kluyveromyces marxianus for a dual microbial bioethanol production system

PONE-D-20-34570R1

Dear Dr. Chieh-Chen Huang,

We’re pleased to inform you that your manuscript has been judged scientifically suitable for publication and will be formally accepted for publication once it meets all outstanding technical requirements.

Kind regards,

Muhammad Aamer Mehmood, Ph.D.

Academic Editor

PLOS ONE

---

## [Editor Report · Acceptance letter]

9 Feb 2021

PONE-D-20-34570R1 

Construction of engineered RuBisCO *Kluyveromyces marxianus* for a dual microbial bioethanol production system 

Dear Dr. Huang:

I'm pleased to inform you that your manuscript has been deemed suitable for publication in PLOS ONE. Congratulations! Your manuscript is now with our production department. 

Kind regards, 

on behalf of

Dr. Muhammad Aamer Mehmood 

Academic Editor

PLOS ONE